# Towards Optimal Multi-draft Speculative Decoding

**Zhengmian Hu**[1,2,*], **Tong Zheng**[1,*], **Vignesh Viswanathan**[2,3], **Ziyi Chen**[1], **Ryan A. Rossi**[2], **Yihan Wu**[1], **Dinesh Manocha**[1,4], **Heng Huang**[1]

[1]Department of Computer Science, University of Maryland, College Park, MD, USA
[2]Adobe Research, San Jose, CA, USA
[3]Manning College of Information & Computer Sciences, University of Massachusetts Amherst, MA, USA
[4]Department of Electrical and Computer Engineering, University of Maryland, College Park, MD, USA

## Abstract

Large Language Models (LLMs) have become an indispensable part of natural language processing tasks. However, autoregressive sampling has become an efficiency bottleneck. Multi-Draft Speculative Decoding (MDSD) is a recent approach where, when generating each token, a small draft model generates multiple drafts, and the target LLM verifies them in parallel, ensuring that the final output conforms to the target model distribution. The two main design choices in MDSD are the draft sampling method and the verification algorithm. For a fixed draft sampling method, the optimal acceptance rate is a solution to an optimal transport problem, but the complexity of this problem makes it difficult to solve for the optimal acceptance rate and measure the gap between existing verification algorithms and the theoretical upper bound. This paper discusses the dual of the optimal transport problem, providing a way to efficiently compute the optimal acceptance rate. For the first time, we measure the theoretical upper bound of MDSD efficiency for vocabulary sizes in the thousands and quantify the gap between existing verification algorithms and this bound. We also compare different draft sampling methods based on their optimal acceptance rates. Our results show that the draft sampling method strongly influences the optimal acceptance rate, with sampling without replacement outperforming sampling with replacement. Additionally, existing verification algorithms do not reach the theoretical upper bound for both without replacement and with replacement sampling. Our findings suggest that carefully designed draft sampling methods can potentially improve the optimal acceptance rate and enable the development of verification algorithms that closely match the theoretical upper bound.

## 1 Introduction

Autoregressive language models have achieved state-of-the-art results in various language tasks (Brown et al., 2020; Touvron et al., 2023), including chatbots (Luo et al., 2022) and code generation (Chen et al., 2021). These models generate outputs by predicting the next token sequentially. However, this autoregressive decoding process leads to significant computational resource requirements and high latency, posing challenges for user experience and limiting potential applications.

Speculative decoding (Leviathan et al., 2023; Chen et al., 2023a) has been proposed to address the high inference cost issue. The method uses a small, fast draft model to generate candidate results, which are then verified and corrected by a large, accurate target model to maintain the model output distribution. Compared to other acceleration methods, such as knowledge distillation, model quantization, and model pruning, speculative decoding has the advantage of significantly reducing inference latency without sacrificing quality of the generated content.

---

[*]Co-first authors.

Multi-Draft Speculative Decoding (MDSD) (Miao et al., 2024; Cai et al., 2024; Li et al., 2024; Spector & Re, 2023) is a recent advancement in speculative decoding. When generating each token, the small draft model generates multiple draft tokens instead of a single one, as in vanilla speculative decoding. The target LLM verifies these tokens in parallel, ensuring that the final output aligns with the target model's distribution while achieving a higher overall acceptance rate than vanilla speculative decoding, as the multiple drafts provide better coverage of the target model's possible outputs.

MDSD algorithms have two main design choices: (1) The draft sampling method. Common approaches include sampling with replacement, where each token is independently sampled from the draft model output distribution, and sampling without replacement, where the probability of selecting a token is updated after each draw to exclude previously selected tokens. (2) The verification algorithm design. Examples include Recursive Rejection Sampling (RRS) (Yang et al., 2024b; Jeon et al., 2024), which sequentially verifies the draft tokens, and K-SEQ (Sun et al., 2024e), which is designed to improve acceptance rate for sampling with replacement.

The acceptance rate, a measure of MDSD algorithm performance, also depends on these two design choices. Any verification algorithm that guarantees the final output aligns with the target model distribution can be viewed as a transport from the draft tokens' distribution to the target model's distribution. For a fixed draft sampling method, the optimal verification algorithm is a solution to an optimal transport problem (Sun et al., 2024e), corresponding to an optimal acceptance rate.

However, the complexity of this optimal transport problem, with the number of variables and constraints growing exponentially with the number of draft tokens, makes it difficult to find efficient solutions. This difficulty has led to two open questions:

(1) For modern LLMs, where the vocabulary size is typically in the thousands, the optimal acceptance rate has never been computed, to the best of our knowledge. Simple linear program (LP) solvers can only compute the optimal transport for small toy models, making it challenging to measure the optimal acceptance rate in practical scenarios.

(2) Although it is widely known that existing verification algorithms are only approximate solutions to the optimal transport problem, the gap between their performance and the theoretical upper bound has never been quantified with respect to real text distribution. Without knowing the optimal acceptance rate, it is difficult to assess how suboptimal these algorithms are.

This paper addresses these two open questions. Our contributions include:

- We transform the problem of solving the optimal acceptance rate corresponding to the optimal transport into a subset selection problem by considering the dual of the problem and then applying total unimodularity. This provides a novel perspective for understanding the efficiency of MDSD.

- For certain special cases, we propose efficient methods to solve the subset selection problem by noticing convexity-like structures in the set function. This includes sampling with replacement and sampling without replacement. For the first time, we provide a practical method to compute the theoretical acceptance rate upper bound of MDSD for a draft distribution.

- For the first time, we measure the theoretical upper bound of MDSD efficiency on real text, and the gap of existing verification algorithms. We compare different draft sampling methods through their optimal acceptance rates and observe that sampling without replacement outperforms sampling with replacement. We evaluate existing verification algorithms, including K-SEQ for with replacement and RRS for without replacement and with replacement sampling, and find that they still have significant gaps from the theoretical upper bound.

- We propose a novel draft sampling method that greedily selects high-probability drafts, with only the last draft being random. In some cases, it achieves an even higher optimal acceptance rate than without replacement. We also propose a corresponding verification algorithm that perfectly reaches the theoretical acceptance rate upper bound.

## 2 PRELIMINARIES

### 2.1 SPECULATIVE DECODING FOR ACCELERATING LLM INFERENCE

Let $\Sigma$ denote the vocabulary set. We have a target model $P_{\text{target}}(\cdot|x_1, x_2, ..., x_m)$, which is a probabilistic model that predicts the probability of the next word. Our goal is to sample from this model as the output.

The process of single step Multi-Draft Speculative Decoding is as follows:

1. For a draft model $P_{\text{draft}}(\cdot|x_1, x_2, ..., x_m)$, sample $n$ draft tokens $\widehat{x}^{(1)}, \ldots \widehat{x}^{(n)}$.
2. Compute the probabilities of the target model in parallel: $P_{\text{target}}(\cdot|x_1, x_2, ..., x_m)$, $P_{\text{target}}(\cdot|x_1, x_2, ..., x_m, \widehat{x}^{(1)})$, ..., $P_{\text{target}}(\cdot|x_1, x_2, ..., x_m, \widehat{x}^{(n)})$. Due to parallel computation, this step is not much slower than computing $P_{\text{target}}(\cdot|x_1, x_2, ..., x_m)$ alone.
3. Run the verification algorithm $x_{m+1} \sim P_{\text{verify}}(\cdot|\widehat{x}^{(1)}, \ldots \widehat{x}^{(n)})$.
4. If accepted, for some draft $\widehat{x}^{(i)}$, we have $x_{m+1} = \widehat{x}^{(i)}$. In this case, we can perform another sampling step $x_{m+2} \sim P_{\text{target}}(\cdot|x_1, x_2, ..., x_m, \widehat{x}^{(i)})$, generating two tokens in one step and achieving acceleration.

Speculative Decoding can generate multiple steps, with multiple drafts at each step and all drafts forming a tree. However, we only consider the single-step case in this paper. For following analysis, we use $p(\cdot) = P_{\text{target}}(\cdot|x_1, x_2, ..., x_m)$ to denote the target distribution and $p_{\text{draft}}$ for distribution of draft tokens.

## 2.2 Speculative Decoding with a Single Draft Token

Informally, the verification algorithm depends on two distributions $p$ and $p_{\text{draft}}$, and one draft token $j \sim p_{\text{draft}}$. The goal is to output $i \sim p$ such that the objective $\max P(i = j)$ is achieved, that is to maximize the probability of random variable $i$ to be the same as random variable $j$.

More formally, given $p \in \Delta_\Sigma$ and $p_{\text{draft}} \in \Delta_\Sigma$ representing two probability distributions over the space $\Sigma$, we seek a joint distribution $\pi \in \Pi(p, p_{\text{draft}})$ such that the marginal distributions are $p$ and $p_{\text{draft}}$, respectively, and the objective $\max \sum_{i \in \Sigma} \pi(i, i)$ is maximized. This forms a optimal transport problem. The optimal transport is denoted as $\pi^*_{p, p_{\text{draft}}} \in \Pi(p, p_{\text{draft}})$, and the optimal objective function value is $\alpha^*(p, p_{\text{draft}}) = \sum_{i \in \Sigma} \pi^*_{p, p_{\text{draft}}}(i, i)$.

The problem can be formulated as an LP by representing the joint distribution as a matrix:

$$\max_{C \in \mathbb{R}^{\Sigma \times \Sigma}} \sum_{i \in \Sigma} C_{i,i} \text{ s.t. } \sum_{j \in \Sigma} C_{i,j} = p(i), \ \sum_{i \in \Sigma} C_{i,j} = p_{\text{draft}}(j), \ C_{i,j} \geq 0 \ \forall i, j \in \Sigma. \tag{1}$$

The optimal transport has the following closed-form expression

$$\pi^*_{p, p_{\text{draft}}}(i, j) = C^*_{i,j} = \begin{cases} \min(p(i), p_{\text{draft}}(i)) & i = j \\ \frac{(p(i) - p_{\text{draft}}(i))_+ (p_{\text{draft}}(j) - p(j))_+}{\sum_{z \in \Sigma}(p_{\text{draft}}(z) - p(z))_+} & i \neq j \end{cases}, \tag{2}$$

and the optimal objective function value is

$$\alpha^*(p, p_{\text{draft}}) = \sum_{i \in \Sigma} \min(p(i), p_{\text{draft}}(i)) . \tag{3}$$

The conditional distribution of $i$ given $j$ when $(i, j) \sim \pi$ is denoted as $\pi(\cdot|j) \in G(\Sigma)$, where

$$\pi(i|j) = \frac{\pi_{i,j}}{\sum_{i \in \Sigma} \pi_{i,j}} = \frac{\pi_{i,j}}{p_{\text{draft}}(j)} . \tag{4}$$

For the optimal transport, this leads to

$$\pi^*_{p, p_{\text{draft}}}(i|j) = \begin{cases} \min(\frac{p(i)}{p_{\text{draft}}(j)}, 1) & i = j \\ (1 - \frac{p(j)}{p_{\text{draft}}(j)})_+ \frac{(p(i) - p_{\text{draft}}(i))_+}{\sum_{z \in \Sigma}(p_{\text{draft}}(z) - p(z))_+} & i \neq j \end{cases}. \tag{5}$$

The basic single-step, single-draft speculative decoding can be improved in two directions. Multi-step methods generate a draft sequence. Some improvements (Sun et al., 2024d; Hu & Huang, 2024; Sun et al., 2024c) in this scenario are discussed in Appendix B.3. Our paper focuses on the multi-draft direction, where multiple draft tokens are generated at each step.

## 2.3 Multi-Draft Speculative Decoding

For $i \in \Sigma$, define the incidence set $A_i := \{\bar{i} \in \Sigma^n | \exists j \in [n], \bar{i}_j = i\}$.

Informally, the verification algorithm depends on two distributions $p$ and $p_{\text{draft}}$, where $p_{\text{draft}}$ is now a joint distribution of $n$ tokens. Common constructions of $p_{\text{draft}}$ include:

- Sampling with replacement: Given a draft model with output distribution $q(\cdot)$, independently sample $n$ times. For $\bar{i} = (\bar{i}_1, \ldots, \bar{i}_n) \in \Sigma^n$, we have $p_{\text{draft}}(\bar{i}) = \prod_{j=1}^n q(\bar{i}_j)$.

- Sampling without replacement: $p_{\text{draft}}(\bar{i}) = \prod_{j=1}^{n} q^{-\bar{i}_1, \ldots, \bar{i}_{j-1}}(\bar{i}_j)$, where

$$q^{-\bar{i}_1, \ldots, \bar{i}_{j-1}}(x) = \begin{cases} \frac{q(x)}{1 - \sum_{z \in \{\bar{i}_1, \ldots, \bar{i}_{j-1}\}} q(z)} & x \notin \{\bar{i}_1, \ldots, \bar{i}_{j-1}\} \\ 0 & x \in \{\bar{i}_1, \ldots, \bar{i}_{j-1}\} \end{cases}. \tag{6}$$

- Product of different draft distributions: $p_{\text{draft}}(\bar{i}) = \prod_{j=1}^{n} q_j(\bar{i}_j)$.

Given multiple draft tokens $\bar{i} = (\bar{i}_1, \ldots, \bar{i}_n) \sim p_{\text{draft}}$, the goal is to output $i \sim p$ such that the objective $\max P(\exists j \in [n], i = \bar{i}_j)$ or equivalently $\max P(\bar{i} \in A_i)$ is achieved, that is to maximize the probability of random variable $i$ to be the same as one of random variable in $(\bar{i}_1, \ldots, \bar{i}_n)$.

More formally, given $p \in \Delta_\Sigma$ and $p_{\text{draft}} \in \Delta_{\Sigma^n}$ representing a probability distribution over the space $\Sigma$ and a probability distribution over the space $\Sigma^n$, respectively, we seek a joint distribution $\pi \in \Pi(p, p_{\text{draft}})$ such that the marginal distributions are $p$ and $p_{\text{draft}}$, respectively, and the objective $\max \sum_{i \in \Sigma} \sum_{\bar{i} \in A_i} \pi(i, \bar{i})$ is maximized. The optimal transport is denoted as $\pi^*_{p, p_{\text{draft}}} \in \Pi(p, p_{\text{draft}})$, and the optimal objective function value is $\alpha^*(p, p_{\text{draft}}) = \sum_{i \in \Sigma} \sum_{\bar{i} \in A_i} \pi^*_{p, p_{\text{draft}}}(i, \bar{i})$.

The problem can be formulated as an LP by representing the joint distribution as a tensor:

$$\max_{C \in \mathbb{R}^{\Sigma \times \Sigma^n}} \sum_{i \in \Sigma} \sum_{\bar{i} \in A_i} C_{i, \bar{i}}$$

$$\text{s.t.} \sum_{\bar{i} \in \Sigma^n} C_{i, \bar{i}} = p(i) \quad \forall i \in \Sigma, \quad \sum_{i \in \Sigma} C_{i, \bar{i}} = p_{\text{draft}}(\bar{i}) \quad \forall \bar{i} \in \Sigma^n, \tag{7}$$

$$C_{i, \bar{i}} \geq 0 \quad \forall i \in \Sigma, \bar{i} \in \Sigma^n.$$

The difficulty lies in the exponential number of variables and constraints.

Several approximation have been proposed for the multi-draft speculative decoding problem, including Recursive Rejection Sampling (RRS) (Yang et al., 2024b; Jeon et al., 2024) and K-SEQ (Sun et al., 2024e). RRS recursively verifies the draft tokens, while K-SEQ improves the acceptance rate for sampling with replacement. Due to space constraints, we move the details of these methods (Appendix A), other related work (Appendix B) and all proofs (Appendix C) to the appendix.

## 3 OPTIMAL ACCEPTANCE RATE AS SUBSET SELECTION PROBLEM

We show that the optimal acceptance rate can be expressed as a subset selection problem:

$$\alpha^*(p, p_{\text{draft}}) = 1 + \min_{H \subset \Sigma} \left( \sum_{i \in H} p(i) - \sum_{\bar{i} \in H^n} p_{\text{draft}}(\bar{i}) \right). \tag{8}$$

### 3.1 DUAL PROBLEM

We start from the linear programming formulation (7) and derive an equivalent formulation:

$$\max_{S \in \mathbb{R}^{\Sigma \times \Sigma^n}} \sum_{i \in \Sigma} \sum_{\bar{i} \in \Sigma^n} S_{i, \bar{i}}$$

$$\text{s.t.} \sum_{\bar{i} \in \Sigma^n} S_{i, \bar{i}} \leq p(i) \quad \forall i \in \Sigma, \quad \sum_{i \in \Sigma} S_{i, \bar{i}} \leq p_{\text{draft}}(\bar{i}) \quad \forall \bar{i} \in \Sigma^n, \tag{9}$$

$$S_{i, \bar{i}} \geq 0 \quad \forall i \in \Sigma, \bar{i} \in \Sigma^n, \quad S_{i, \bar{i}} = 0 \quad \forall i \in \Sigma, \bar{i} \notin A_i.$$

**Lemma 1.** *The two formulations* (7) *and* (9) *are equivalent.*

This equivalent formulation transforms the transportation problem (Hitchcock, 1941) into a $b$-matching problem, whose dual is a $w$-vertex cover problem (Schrijver et al., 2003) (with a detailed derivation in Appendix C.1):

$$\min_{y \in \mathbb{R}^\Sigma, z \in \mathbb{R}^{\Sigma^n}} \sum_{i \in \Sigma} y_i p(i) + \sum_{\bar{i} \in \Sigma^n} z_{\bar{i}} p_{\text{draft}}(\bar{i})$$

$$\text{s.t. } y_i + z_{\bar{i}} \geq 1 \quad \forall i \in \Sigma, \bar{i} \in A_i, \quad y_i \geq 0 \quad \forall i \in \Sigma, \quad z_{\bar{i}} \geq 0 \quad \forall \bar{i} \in \Sigma^n. \tag{10}$$

### 3.2 TOTAL UNIMODULARITY

The coefficient matrix of the constraints in (10) is totally unimodular (TUM). The first set of constraints forms an incidence matrix of a bipartite graph, where one side of the nodes corresponds to $\Sigma$ and the other side corresponds to $\Sigma^n$. There is an edge between $i$ and $\bar{i}$ if and only if $\bar{i} \in A_i$. Therefore, it is a totally unimodular matrix (Biggs, 1993). The second and third sets of constraints have

coefficient matrices that are identity matrices, with $|\Sigma| + |\Sigma|^n$ variables and $|\Sigma| + |\Sigma|^n$ constraints. The concatenation of a TUM matrix and an identity matrix is also TUM (Commoner, 1973).

Since the right-hand side of the constraints are integers, the dual problem (10) always has an integer optimal solution (Hoffman & Kruskal, 2010).

### 3.3 SUBSET SELECTION FORMULATION

By restricting the variables in (10) to integers, we obtain:

$$\min_{y \in \mathbb{Z}^\Sigma, z \in \mathbb{Z}^{\Sigma^n}} \sum_{i \in \Sigma} y_i p(i) + \sum_{\bar{i} \in \Sigma^n} z_{\bar{i}} p_{\text{draft}}(\bar{i}) \tag{11}$$
$$\text{s.t. } y_i + z_{\bar{i}} \geq 1 \quad \forall i \in \Sigma, \bar{i} \in A_i, \quad y_i \geq 0 \quad \forall i \in \Sigma, \quad z_{\bar{i}} \geq 0 \quad \forall \bar{i} \in \Sigma^n.$$

In the optimal solution, $y_i$ and $z_{\bar{i}}$ will not exceed 1, so they can only take values 0 or 1. Therefore, the problem can be further simplified as:

$$\min_{y \in \{0,1\}^\Sigma} \min_{z \in \{0,1\}^{\Sigma^n}} \sum_{i \in \Sigma} y_i p(i) + \sum_{\bar{i} \in \Sigma^n} z_{\bar{i}} p_{\text{draft}}(\bar{i}) \tag{12}$$
$$\text{s.t. } y_i + z_{\bar{i}} \geq 1 \quad \forall i \in \Sigma, \bar{i} \in A_i .$$

Define $H = \{i \in \Sigma | y_i = 1\}$. The problem becomes:

$$\min_{H \subset \Sigma} \min_{z \in \{0,1\}^{\Sigma^n}} \sum_{i \in H} p(i) + \sum_{\bar{i} \in \Sigma^n} z_{\bar{i}} p_{\text{draft}}(\bar{i}) \tag{13}$$
$$\text{s.t. } z_{\bar{i}} \geq 1 \quad \forall i \in \Sigma \setminus H, \bar{i} \in A_i .$$

The optimal solution for $z$ is

$$z^*(H)_{\bar{i}} = \begin{cases} 1 & \bar{i} \in \bigcup_{x \in \Sigma \setminus H} A_x \\ 0 & \bar{i} \notin \bigcup_{x \in \Sigma \setminus H} A_x \end{cases} . \tag{14}$$

Substituting this solution, we obtain the subset selection formulation:

$$\min_{H \subset \Sigma} \sum_{i \in H} p(i) + \sum_{\bar{i} \in \bigcup_{x \in \Sigma \setminus H} A_x} p_{\text{draft}}(\bar{i}) . \tag{15}$$

Finally, note that

$$\sum_{\bar{i} \in \bigcup_{x \in \Sigma \setminus H} A_x} p_{\text{draft}}(\bar{i}) + \sum_{\bar{i} \in H^n} p_{\text{draft}}(\bar{i}) = 1 . \tag{16}$$

This completes the derivation of the subset selection formulation (8).

## 4 COMPUTING OPTIMAL ACCEPTANCE RATE IN SPECIAL CASES

In this section, we discuss how to efficiently compute the optimal acceptance rate for certain special cases of the draft distribution $p_{\text{draft}}$. For any set function $f$, we define the marginal value of an element $x$ with respect to a set $H$ as $f(x|H) = f(H \cup \{x\}) - f(H)$. We also define the following shorthand notations: $P(H) = \sum_{i \in H} p(i)$, $Q(H) = \sum_{\bar{i} \in H^n} p_{\text{draft}}(\bar{i})$, $f(H) = P(H) - Q(H)$.

The optimal acceptance rate can be expressed as $\alpha^*(p, p_{\text{draft}}) = 1 + \min_{H \subset \Sigma} f(H)$.

### 4.1 $q$-CONVEX FUNCTIONS

**Definition 2** ($q$-Convex Function). *A set function $Q : 2^\Sigma \to \mathbb{R}$ is called a q-convex function if there exists a function $q : \Sigma \to \mathbb{R}_{\geq 0}$ such that for all $H \subset \Sigma$ and $x, y \in \Sigma \setminus H$ with $x \neq y$, we have*

$$\frac{Q(x|H)}{q(x)} \leq \frac{Q(y|H \cup \{x\})}{q(y)} . \tag{17}$$

Intuitively, if we order the elements of $\Sigma$ arbitrarily and construct a sequence of sets $H_i$ by adding elements one by one, then the curve of $Q(H_i)$ against the sum of $q$ values is always convex.

**Theorem 3.** *For sampling with replacement, the function $Q$ is a q-convex function.*

**Theorem 4.** *For sampling without replacement, the function $Q$ is a q-convex function.*

**Theorem 5.** *All q-convex functions are supermodular functions.*

For both sampling with replacement and without replacement, computing $\alpha^*(p, p_{\text{draft}})$ can be formulated as an unconstrained submodular minimization problem, which has polynomial-time algorithms (Iwata, 2008). However, by fully exploiting the properties of $q$-convex functions, we can solve the problem even faster, as shown in the next section.

## 4.2 EFFICIENT COMPUTATION

**Theorem 6.** *Suppose that $Q$ is a q-convex function, $Q$ is monotone increasing, and $p(x) > 0$ for all $x \in \Sigma$. For all $H \subset \Sigma$ and $x, y \in \Sigma \setminus H$ with $x \neq y$, if $\frac{q(x)}{p(x)} \leq \frac{q(y)}{p(y)}$ and $f(x|H) \leq 0$, then $f(y|H \cup \{x\}) \leq 0$.*

The above theorem requires $p(x) > 0$. When $p(x) = 0$, there exists an optimal set $H^*$ for (8) that contains $x$ because $Q$ is monotone increasing.

### 4.2.1 ALGORITHM

Inspired by Theorem 6, we can compute the optimal acceptance rate efficiently as follows:

1. Find an ordering $\sigma$ of $\Sigma$ such that $\frac{q(\sigma_1)}{p(\sigma_1)} \geq \cdots \geq \frac{q(\sigma_{|\Sigma|})}{p(\sigma_{|\Sigma|})}$.
2. Construct a sequence of sets $H_i = \{\sigma_1, \ldots, \sigma_i\}$.
3. Compute $\alpha^*(p, p_{\text{draft}}) = 1 + \min_i f(H_i)$.

Intuitively, we sort the elements by the ratio of $q$ and $p$ in non-increasing order and then perform a linear search.

### 4.2.2 COMPLEXITY OF COMPUTING $Q$ AND $\alpha^*$

For sampling with replacement, $Q$ has a simple expression $Q(H) = (\sum_{x \in H} q(x))^n$. The time complexity for computing $\alpha^*(p, p_{\text{draft}})$ is $O(|\Sigma| \log |\Sigma|)$ for the sorting step, plus $O(|\Sigma|)$ for the linear scan.

For sampling without replacement, we can compute $Q(H) = \frac{W_{n,H}}{W_{n,\Sigma}}$ based on the coefficient of generating function $W_{n,H} = \text{Coeff}_{t^n} G_H(t) = \text{Coeff}_{t^n} \prod_{i \in H}(1 + q(i)t)$ and apply dynamic programming with recurrence relation $W_{n,H \cup \{x\}} = W_{n,H} + q(x)W_{n-1,H}$. The time complexity is $O(|\Sigma| \log |\Sigma|)$ for the sorting step, plus $O(n|\Sigma|)$ for computing coefficient of generating function with dynamic programming.

## 5 A GREEDY APPROACH FOR SELECTING DRAFT TOKENS

In this section, we propose a novel method for constructing the draft distribution $p_{\text{draft}}$ and a corresponding verification algorithm that achieves the optimal acceptance rate for this distribution.

### 5.1 DRAFT CONSTRUCTION

Given a draft model output distribution $q \in \Delta_\Sigma$, we construct the draft tokens $\bar{i} = (\bar{i}_1, \ldots, \bar{i}_n)$ as follows:

- The first $n - 1$ tokens are deterministically set to be the top $n - 1$ tokens according to the probability in $q$, i.e., $\bar{i}_1, \ldots, \bar{i}_{n-1} = \text{Top}_{n-1}(q)$, such that $q(\bar{i}_1) \geq \cdots \geq q(\bar{i}_{n-1})$ and $\max_{i \in \Sigma \setminus \{\bar{i}_1, \ldots, \bar{i}_{n-1}\}} q(i) \leq q(\bar{i}_{n-1})$.
- Only the last token $\bar{i}_n$ is randomly sampled from $q$ without replacement (i.e., it is different from the previous $n - 1$ tokens): $\bar{i}_n \sim q^{\neg \text{Top}_{n-1}(q)}(\cdot) = \frac{q(\cdot)}{1 - \sum_{j=1}^{n-1} q(\bar{i}_j)}$.

The resulting draft distribution is

$$p_{\text{draft}}(\bar{i}) = \begin{cases} q^{\neg \text{Top}_{n-1}(q)}(\bar{i}_n) & \bar{i}_1, \ldots, \bar{i}_{n-1} = \text{Top}_{n-1}(q) \\ 0 & \bar{i}_1, \ldots, \bar{i}_{n-1} \neq \text{Top}_{n-1}(q) \end{cases}. \tag{18}$$

## 5.2 Verification Algorithm

The corresponding optimal transport problem for this draft distribution is simple because only one draft token is random. We can design a verification algorithm that strictly achieves the optimal acceptance rate for this draft distribution (, with unfolded definition in Appendix D):

$$\pi_{p,p_{\text{draft}}}^{\text{Greedy}}(i|\bar{i}) = \pi_{p,q^{\neg \text{Top}_{n-1}(q)}}^*(i|\bar{i}_n) \,. \tag{19}$$

**Theorem 7.** *The optimal acceptance rate for the greedy draft distribution is*

$$\alpha^*(p, p_{\text{draft}}) = \alpha^{\text{Greedy}}(p, p_{\text{draft}}) = \sum_{i \in \text{Top}_{n-1}(q)} p(i) + \sum_{i \in \Sigma} \min(p(i), q^{\neg \text{Top}_{n-1}(q)}(i)) \,. \tag{20}$$

Our subset selection formulation (8) provides a convenient way to prove the above theorem.

## 5.3 Connection to SpecHub

SpecHub (Sun et al., 2024b) is a recently proposed MDSD method that is only applicable to the case of $n = 2$. The draft construction in SpecHub is as follows:

- First, sample the first draft token $\bar{i}_1$.
- If $\bar{i}_1 = \text{Top}_1(q)$ is the token with the highest probability in $q$, then sample the second draft token $\bar{i}_2$ without replacement to ensure it is different from $\bar{i}_1$.
- If $\bar{i}_1 \neq \text{Top}_1(q)$ is not the token with the highest probability in $q$, then deterministically set the second draft token to be the token with the highest probability, i.e., $\bar{i}_2 = \text{Top}_1(q)$.

The resulting draft distribution is:

$$p_{\text{draft}}(\bar{i}) = \begin{cases} q(\bar{i}_1) & \bar{i}_2 = \text{Top}_1(q) \\ \frac{q(\bar{i}_1)}{1 - q(\bar{i}_1)} q(\bar{i}_2) & \bar{i}_1 = \text{Top}_1(q) \\ 0 & \text{otherwise} \end{cases} \,. \tag{21}$$

We note that the greedy method for $n = 2$ is essentially equivalent to SpecHub because both methods ensure that at least one draft token is the token with the highest probability in $q$. However, the specific draft distributions are different, leading to a simpler verification algorithm for the greedy method.

## 6 Experiments

The goal of our experiments is to measure the acceptance rates of various MDSD methods on real text distributions and compare them with the theoretical upper bounds. In the previous sections, we analyzed the theoretical acceptance rate $\alpha^*(p, p_{\text{draft}})$ for three different draft distributions: sampling with replacement, sampling without replacement, and greedy approach (Section 5). We also discussed some existing verification methods (Appendix A), such as RRS and K-SEQ, whose acceptance rates are expected to be lower than the theoretical upper bound. For K-SEQ, its average acceptance rate $\alpha^{\text{K-SEQ}}$ can be derived theoretically (see Appendix A.2 for details). Our efficient computation methods (Section 4) make it possible, for the first time, to obtain the theoretical upper bound of MDSD for vocabulary sizes of thousands.

To obtain realistic distributions $p$ and $p_{\text{draft}}$, we select real-world datasets for various tasks, including Alpaca (Taori et al., 2023) for instruction-following, WMT'14 De-En (Bojar et al., 2014) for translation, and CNN-DailyMail (Hermann et al., 2015) for summarization. For each task, we use an LLM to generate responses on 1024 data samples, with a maximum length of 128 tokens. We then measure the logits of the target model and the draft model on these generated responses to construct $p$ and $p_{\text{draft}}$.

We evaluated different approaches based on four publicly available large language models, including 1) LLaMA (Touvron et al., 2023), 2) Vicuna (Chiang et al., 2023), the instruction fine-tuned version of LLaMA models, 3) OPT (Zhang et al., 2022), and 4) Qwen2 (Yang et al., 2024a). Specifically, for the LLaMA family, we select LLaMA-7B as the target model and LLaMA-68M as the draft model, which is consistent with previous work (Miao et al., 2024). For the OPT family, we select OPT-6.7B as the target model and OPT-125M as the draft model. Moreover, for the Vicuna family

Table 1: Acceptance rates of different MDSD methods across various models and tasks. $\Delta\alpha$ means the gap between a verification method and the theoretical upper bound, with statistically significant differences indicated by directional arrows.

| Model Pairs | Draft Sampling | Method | Alpaca | | CNN-DailyMail | | WMT'14 | |
|---|---|---|---|---|---|---|---|---|
| | | | $\alpha$ | $\Delta\alpha$ | $\alpha$ | $\Delta\alpha$ | $\alpha$ | $\Delta\alpha$ |
| OPT-125M OPT-6.7B | With Replacement | RRS | $85.4 \pm 0.1$ | $-1.5\downarrow$ | $77.3 \pm 0.1$ | $-3.3\downarrow$ | $70.6 \pm 0.1$ | $-1.5\downarrow$ |
| | | K-SEQ | $85.8 \pm 0.1$ | $-1.1\downarrow$ | $78.4 \pm 0.1$ | $-2.2\downarrow$ | $71.1 \pm 0.1$ | $-0.9\downarrow$ |
| | | $\alpha^{\text{K-SEQ}}$ | $85.9 \pm 0.1$ | $-0.9\downarrow$ | $78.5 \pm 0.1$ | $-2.2\downarrow$ | $71.0 \pm 0.1$ | $-1.0\downarrow$ |
| | | $\alpha^*$ | $\mathbf{86.9 \pm 0.1}$ | - | $\mathbf{80.7 \pm 0.1}$ | - | $\mathbf{72.0 \pm 0.1}$ | - |
| | Without Replacement | RRS | $88.9 \pm 0.1$ | $-0.9\downarrow$ | $81.5 \pm 0.1$ | $-2.8\downarrow$ | $75.1 \pm 0.1$ | $-1.0\downarrow$ |
| | | $\alpha^*$ | $\mathbf{89.9 \pm 0.1}$ | - | $\mathbf{84.3 \pm 0.1}$ | - | $\mathbf{76.0 \pm 0.1}$ | - |
| | Greedy | Verify | $\mathbf{90.7 \pm 0.1}$ | $0.0$ | $84.2 \pm 0.1$ | $-0.1$ | $\mathbf{77.0 \pm 0.1}$ | $-0.0$ |
| | | $\alpha^*$ | $\mathbf{90.7 \pm 0.1}$ | - | $\mathbf{84.3 \pm 0.1}$ | - | $\mathbf{77.1 \pm 0.1}$ | - |
| LLaMA-68M LLaMA-7B | With Replacement | RRS | $71.6 \pm 0.1$ | $-1.5\downarrow$ | $65.3 \pm 0.1$ | $-2.2\downarrow$ | $59.8 \pm 0.1$ | $-1.0\downarrow$ |
| | | K-SEQ | $71.9 \pm 0.1$ | $-1.1\downarrow$ | $66.0 \pm 0.1$ | $-1.5\downarrow$ | $60.0 \pm 0.1$ | $-0.8\downarrow$ |
| | | $\alpha^{\text{K-SEQ}}$ | $72.0 \pm 0.1$ | $-1.0\downarrow$ | $66.2 \pm 0.1$ | $-1.3\downarrow$ | $60.2 \pm 0.1$ | $-0.6\downarrow$ |
| | | $\alpha^*$ | $\mathbf{73.0 \pm 0.1}$ | - | $\mathbf{67.5 \pm 0.1}$ | - | $\mathbf{60.8 \pm 0.1}$ | - |
| | Without Replacement | RRS | $75.7 \pm 0.1$ | $-0.8\downarrow$ | $70.5 \pm 0.1$ | $-1.3\downarrow$ | $\mathbf{63.3 \pm 0.1}$ | $-0.1$ |
| | | $\alpha^*$ | $\mathbf{76.5 \pm 0.1}$ | - | $\mathbf{71.8 \pm 0.1}$ | - | $\mathbf{63.4 \pm 0.1}$ | - |
| | Greedy | Verify | $\mathbf{78.4 \pm 0.1}$ | $-0.1$ | $\mathbf{73.2 \pm 0.1}$ | $0.1$ | $\mathbf{66.1 \pm 0.1}$ | $0.0$ |
| | | $\alpha^*$ | $\mathbf{78.4 \pm 0.1}$ | - | $\mathbf{73.1 \pm 0.1}$ | - | $\mathbf{66.1 \pm 0.1}$ | - |
| Eagle-0.24B Vicuna-7B | With Replacement | RRS | $63.4 \pm 0.2$ | $-1.0\downarrow$ | $56.7 \pm 0.1$ | $-1.1\downarrow$ | $32.9 \pm 0.2$ | $-0.2$ |
| | | K-SEQ | $63.9 \pm 0.2$ | $-0.5\downarrow$ | $57.0 \pm 0.1$ | $-0.8\downarrow$ | $33.0 \pm 0.2$ | $-0.1$ |
| | | $\alpha^{\text{K-SEQ}}$ | $63.7 \pm 0.1$ | $-0.7\downarrow$ | $57.1 \pm 0.1$ | $-0.7\downarrow$ | $32.9 \pm 0.2$ | $-0.1$ |
| | | $\alpha^*$ | $\mathbf{64.4 \pm 0.1}$ | - | $\mathbf{57.8 \pm 0.1}$ | - | $\mathbf{33.1 \pm 0.2}$ | - |
| | Without Replacement | RRS | $70.9 \pm 0.2$ | $-0.6\downarrow$ | $63.4 \pm 0.1$ | $-0.8\downarrow$ | $\mathbf{36.9 \pm 0.2}$ | $0.4$ |
| | | $\alpha^*$ | $\mathbf{71.5 \pm 0.1}$ | - | $\mathbf{64.2 \pm 0.1}$ | - | $36.4 \pm 0.2$ | - |
| | Greedy | Verify | $\mathbf{72.6 \pm 0.2}$ | $-0.2$ | $\mathbf{65.7 \pm 0.1}$ | $-0.1$ | $\mathbf{39.5 \pm 0.2}$ | $0.1$ |
| | | $\alpha^*$ | $\mathbf{72.8 \pm 0.1}$ | - | $\mathbf{65.8 \pm 0.1}$ | - | $\mathbf{39.5 \pm 0.2}$ | - |
| Eagle-0.26B Qwen2-7B | With Replacement | RRS | $59.6 \pm 0.2$ | $-1.0\downarrow$ | $46.7 \pm 0.1$ | $-1.6\downarrow$ | $38.3 \pm 0.1$ | $-0.4\downarrow$ |
| | | K-SEQ | $59.9 \pm 0.2$ | $-0.8\downarrow$ | $47.2 \pm 0.1$ | $-1.1\downarrow$ | $\mathbf{38.3 \pm 0.1}$ | $-0.4$ |
| | | $\alpha^{\text{K-SEQ}}$ | $59.9 \pm 0.1$ | $-0.7\downarrow$ | $47.3 \pm 0.1$ | $-1.0\downarrow$ | $\mathbf{38.3 \pm 0.1}$ | $-0.3$ |
| | | $\alpha^*$ | $\mathbf{60.7 \pm 0.1}$ | - | $\mathbf{48.3 \pm 0.1}$ | - | $\mathbf{38.7 \pm 0.1}$ | - |
| | Without Replacement | RRS | $68.3 \pm 0.2$ | $-1.1\downarrow$ | $52.4 \pm 0.1$ | $-1.7\downarrow$ | $\mathbf{43.9 \pm 0.1}$ | $-0.1$ |
| | | $\alpha^*$ | $\mathbf{69.4 \pm 0.1}$ | - | $\mathbf{54.1 \pm 0.1}$ | - | $\mathbf{44.0 \pm 0.1}$ | - |
| | Greedy | Verify | $\mathbf{69.9 \pm 0.2}$ | $-0.0$ | $\mathbf{54.0 \pm 0.1}$ | $0.0$ | $\mathbf{45.5 \pm 0.1}$ | $0.1$ |
| | | $\alpha^*$ | $\mathbf{70.0 \pm 0.1}$ | - | $\mathbf{53.9 \pm 0.1}$ | - | $\mathbf{45.4 \pm 0.1}$ | - |

and the Qwen family, we select Vicuna-7B-v1.3 and Qwen2-7B-Instruct as target models, and we use paired draft models provided by EAGEL (Li et al., 2024), with 0.24B parameters and 0.26B parameters, respectively.

Unless otherwise specified, we use a default generation temperature of 0.7 and a draft token number of 3. The total computational cost is less than 50 GPU hours on RTXA6000.

## 6.1 MAIN EXPERIMENT

In the main experiment, we compare the acceptance rates of different MDSD methods across various LLMs and tasks. The results are shown in Table 1. We observe that the existing verify methods, RRS and K-SEQ, still have gaps compared to the theoretical acceptance rate upper bound. Sampling without replacement achieves higher acceptance rates than sampling with replacement, both in terms of the theoretical upper bound and the existing verification algorithms. We can attribute this to the fact that sampling with replacement may lead to duplicate draft tokens, which are less helpful for acceleration. The greedy method obtains the highest acceptance rate, but this is not always the case, as we will see in the ablation study below that the greedy method performs worse when the temperature is 1.

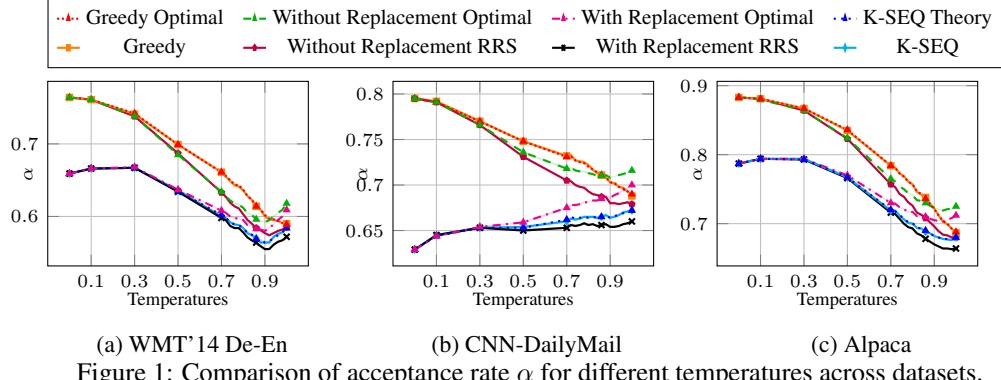

(a) WMT'14 De-En     (b) CNN-DailyMail     (c) Alpaca

Figure 1: Comparison of acceptance rate $\alpha$ for different temperatures across datasets.

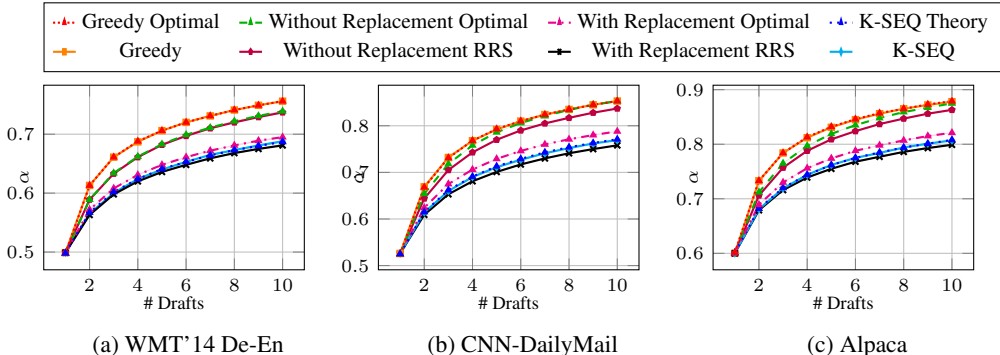

(a) WMT'14 De-En     (b) CNN-DailyMail     (c) Alpaca

Figure 2: Comparison of acceptance rate $\alpha$ for different number of drafts across datasets.

## 6.2 ABLATION STUDY I: IMPACT OF TEMPERATURE

We study the impact of different temperatures on the acceptance rates. The temperature affects the distributions of the target model and the draft model, even if the logits remain unchanged. It also affects the output text during the sampling process, resulting in different responses. Figure 1 shows the results. We use LLaMA-7B as the target model and LLaMA-68M as the draft model for our ablation studies. We can have the following observations:

- The impact of temperature is non-monotonic. Moreover, different methods respond differently to temperature changes.
- At low temperatures, all methods fall into two categories. The first includes methods that allow duplicate tokens. When $T = 0$, these methods essentially have only one effective draft token, the one with the largest logits on the draft model. The second includes methods that prevent duplicate tokens. When $T = 0$, these methods always select the top $n$ tokens on the draft model.
- The gap between the optimal acceptance rate and acceptance rates for previously existing verification methods, RRS and K-SEQ, gradually increases as the temperature rises.
- As temperature increases, the gap between methods with replacement sampling and methods without replacement sampling decreases. We can attribute this to the fact that, at high temperatures, the probability distribution is less concentrated, making with replacement sampling strategies have less probability to generate duplicate tokens.

## 6.3 ABLATION STUDY II: IMPACT OF NUMBER OF DRAFTS

We investigate the impact of different numbers of drafts on the acceptance rates. The results are shown in Figure 2. We have the following observations:

- As the number of drafts increases, the coverage of the target model's possible outputs improves, therefore leading to better acceptance rate. This trend holds for all methods.
- The draft sampling strategy significantly impacts the benefits derived from an increase in the number of drafts. Sampling without replacement generally benefit more from an increase in drafts compared to sampling with replacement. This is because sampling with replacement can lead to redundant drafts, which do not fully leverage the advantages of increasing the number of drafts.

Table 2: Acceptance rates of different MDSD methods on MT-Bench based on Eagle framework.

| Method | # Drafts = 2, # Steps = 4 | | # Drafts = 4, # Steps = 3 | | EAGLE default sparse tree | |
|---|---|---|---|---|---|---|
| | $\alpha$ | Speed | $\alpha$ | Speed | $\alpha$ | Speed |
| *T = 0.1* | | | | | | |
| RRS w/ replacement | 75.3 ± 0.3 | - | 78.4 ± 0.3 | - | 74.7 ± 0.3 | - |
| RRS w/o replacement | 79.4 ± 0.3 | 1.04 (± 0.02) × | 80.4 ± 0.3 | 1.03 (± 0.01) × | 76.8 ± 0.3 | 1.04 (± 0.02) × |
| SpecHub | **84.0** ± 0.3 | 1.11 (± 0.02) × | - | - | - | - |
| Greedy | **84.7** ± 0.3 | 1.13 (± 0.02) × | **88.8** ± 0.2 | 1.17 (± 0.01) × | **79.1** ± 0.3 | 1.08 (± 0.02) × |
| *T = 0.6* | | | | | | |
| RRS w/ replacement | 78.8 ± 0.3 | - | 84.7 ± 0.3 | - | 76.2 ± 0.3 | - |
| RRS w/o replacement | **82.4** ± 0.3 | 1.07 (± 0.02) × | 88.6 ± 0.2 | 1.07 (± 0.01) × | **77.6** ± 0.3 | 1.05 (± 0.02) × |
| SpecHub | 82.3 ± 0.3 | 1.02 (± 0.02) × | - | - | - | - |
| Greedy | **82.8** ± 0.3 | 1.04 (± 0.02) × | **90.0** ± 0.2 | 1.09 (± 0.01) × | **78.3** ± 0.3 | 1.01 (± 0.02) × |
| *T = 1.0* | | | | | | |
| RRS w/ replacement | 76.7 ± 0.3 | - | 83.5 ± 0.3 | - | 72.1 ± 0.3 | - |
| RRS w/o replacement | 76.4 ± 0.3 | 1.00 (± 0.02) × | 85.3 ± 0.3 | 1.05 (± 0.01) × | **74.1** ± 0.3 | 1.03 (± 0.02) × |
| SpecHub | **79.5** ± 0.3 | 1.01 (± 0.02) × | - | - | - | - |
| Greedy | **79.2** ± 0.3 | 1.02 (± 0.02) × | **87.8** ± 0.2 | 1.08 (± 0.01) × | 72.9 ± 0.3 | 0.97 (± 0.02) × |

- The gap between the optimal acceptance rate and acceptance rates for previously existing verification methods, RRS and K-SEQ, gradually increases as the number of drafts rises.

## 6.4 Evaluating the Greedy Sampling Method on Generation Tasks

In this section, we evaluate the effectiveness and generation efficiency of the proposed Greedy draft sampling method (Section 5) on real-world generation tasks and compare it with other MDSD methods.

We implement the Greedy method within the EAGLE Framework (Li et al., 2024), which supports multi-step MDSD with a draft tree structure. We experiment with three types of tree structures: (1) drafts = 2, depths = 4; (2) drafts = 4, depths = 3; and (3) a sparse tree with up to 4 drafts and 5 steps, which is the default setting in EAGLE. We conduct experiments on the MT-Bench dataset (Zheng et al., 2023) using Vicuna-7B-v1.3 (Chiang et al., 2023) as the target model and its corresponding Eagle model with 0.24B parameters as the draft model.

Table 3 presents the results. As discussed in Section 5.3, the Greedy method and SpecHub have equal acceptance rates when the number of draft tokens is 2. Our experiments confirm this theoretical insight, showing no statistically significant difference between the two methods for any temperature.

The Greedy method demonstrates improved performance at low temperatures. For example, at T=0.1, it achieves a higher acceptance rate compared to RRS without replacement, leading to faster generation. However, as the temperature increases, the performance gain of the Greedy method diminishes. This observation is consistent with the ablation study in Figure 1.

## 7 Conclusion

In this paper, we studied the acceptance rate of Multi-Draft Speculative Decoding (MDSD).

On the theoretical side, we discovered an equivalence between the optimal acceptance rate and a subset selection problem. We also provided efficient methods to compute the optimal acceptance rate for common draft distributions.

On the practical side, for the first time, we measured the optimal acceptance rate under real text distributions and quantified the gap between existing algorithms and the optimal acceptance rate.

Furthermore, we proposed a practical greedy draft construction method that, in some cases, achieves an even higher acceptance rate than sampling without replacement.

We hope that our work will stimulate further research on improving the efficiency of large language model inference and make these powerful models more accessible and applicable in real-world scenarios.

ACKNOWLEDGMENT

This work was partially supported by NSF IIS 2347592, 2348169, DBI 2405416, CCF 2348306, CNS 2347617.

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

# A   APPROXIMATE SOLUTIONS

## A.1   RECURSIVE REJECTION SAMPLING (RRS)

Yang et al. (2024b) and Jeon et al. (2024) use the Recursive Rejection Sampling method. Define the residual distribution $\text{Res}^{p-q} \in \Delta_\Sigma$ for $p, q \in \Delta_\Sigma$ as:

$$\text{Res}^{p-q}(i) = \frac{(p(i) - q(i))_+}{\sum_{z \in \Sigma}(p(z) - q(z))_+} \tag{22}$$

When $p = q$, $\text{Res}^{p-q}$ can be defined as an arbitrary distribution.

For $p_{\text{draft}}$ from sampling with replacement, the RRS algorithm is recursively defined as:

$$\pi_{p,p_{\text{draft}}}^{\text{RRS,w}}(i|\bar{i}) = \widetilde{\pi}_{p,q}^{\text{RRS,w}}(i|\bar{i}) \tag{23}$$

where

$$\widetilde{\pi}_{p,q}^{\text{RRS,w}}(i|\bar{i}) = \begin{cases} \min(\frac{p(\bar{i}_1)}{q(\bar{i}_1)}, 1) & i = \bar{i}_1 \\ (1 - \frac{p(\bar{i}_1)}{q(\bar{i}_1)})_+ \widetilde{\pi}_{\text{Res}^{p-q},q}^{\text{RRS,w}}(i|\bar{i}_{2:}) & i \neq \bar{i}_1 \end{cases} \tag{24}$$

and

$$\widetilde{\pi}_{p,q}^{\text{RRS,w}}(i|()) = p(i) \tag{25}$$

Here $\bar{i}_{2:}$ denotes the sequence $\bar{i}$ with the first element removed.

For $p_{\text{draft}}$ from sampling without replacement, the RRS algorithm is defined as:

$$\pi_{p,p_{\text{draft}}}^{\text{RRS,w}}(i|\bar{i}) = \widetilde{\pi}_{p,q}^{\text{RRS,wo}}(i|\bar{i}) \tag{26}$$

where

$$\widetilde{\pi}_{p,q}^{\text{RRS,wo}}(i|\bar{i}) = \begin{cases} \min(\frac{p(\bar{i}_1)}{q(\bar{i}_1)}, 1) & i = \bar{i}_1 \\ (1 - \frac{p(\bar{i}_1)}{q(\bar{i}_1)})_+ \widetilde{\pi}_{\text{Res}^{p-q},q^{\neg\bar{i}_1}}^{\text{RRS,wo}}(i|\bar{i}_{2:}) & i \neq \bar{i}_1 \end{cases} \tag{27}$$

and

$$\widetilde{\pi}_{p,q}^{\text{RRS,wo}}(i|()) = p(i) \tag{28}$$

The acceptance rates are denoted as $\alpha^{\text{RRS,w}}(p, p_{\text{draft}})$ and $\alpha^{\text{RRS,wo}}(p, p_{\text{draft}})$, respectively.

## A.2   K-SEQ

Sun et al. (2024e) proposed the K-SEQ method to verify drafts sampled with replacement. Define

$$\beta_{p,q}(\rho) = \sum_{i \in \Sigma} \min(\frac{p(i)}{\rho}, q(i)) \tag{29}$$

and let $\rho$ be the solution to the equation

$$1 - (1 - \beta_{p,q}(\rho))^n = \rho \beta_{p,q}(\rho) \tag{30}$$

The K-SEQ algorithm is defined as:

$$\pi_{p,p_{\text{draft}}}^{\text{K-SEQ}}(i|\bar{i}) = \widetilde{\pi}_{p,q,\rho}^{\text{K-SEQ}}(i|\bar{i}) \tag{31}$$

where

$$\widetilde{\pi}_{p,q,\rho}^{\text{K-SEQ}}(i|\bar{i}) = (1 - \frac{p(\bar{i}_1)}{\rho q(\bar{i}_1)})_+ \widetilde{\pi}_{p,q,\rho}^{\text{K-SEQ}}(i|\bar{i}_{2:}) + \begin{cases} \min(\frac{p(\bar{i}_1)}{\rho q(\bar{i}_1)}, 1) & i = \bar{i}_1 \\ 0 & i \neq \bar{i}_1 \end{cases} \tag{32}$$

and

$$\widetilde{\pi}_{p,q,\rho}^{\text{K-SEQ}}(i|()) = \frac{p(i) - \min\left\{q(i), \frac{p(i)}{\rho}\right\}\frac{1-(1-\beta_{p,q}(\rho))^n}{\beta_{p,q}(\rho)}}{(1 - \beta_{p,q}(\rho))^n} \tag{33}$$

The acceptance rate is denoted as $\alpha^{\text{K-SEQ}}(p, p_{\text{draft}}) = 1 - (1 - \beta_{p,q}(\rho))^n$, which is theoretically guaranteed to achieve a $(1 - e^{-1})$-approximation of the optimal acceptance rate.

# B  RELATED WORKS

## B.1  DRAFT MODEL DESIGN

Numerous studies have explored the design of better draft models for speculative decoding. In principle, any autoregressive probabilistic model can serve as a draft model. The simplest approaches include using n-gram models (Ou et al., 2024) or document retrieval as draft models (Yang et al., 2023; He et al., 2023). Small transformer-based language models have also been employed (Leviathan et al., 2023; Chen et al., 2023a), often with distillation techniques to further increase the overlap between the draft and target models (Zhou et al., 2023).

The design of a good draft model involves a trade-off between its similarity to the target model and its computational complexity. More complex draft models lead to higher acceptance rates due to their closer resemblance to the target model, but they also incur higher computational overhead. To achieve a better trade-off, some works have proposed reusing the target model's computational results. For example, Monea et al. (2023) use the original model with "look ahead" tokens, while Cai et al. (2024) add new heads to the last hidden layer of the original model to predict tokens further ahead. Li et al. (2024) reuse the last layer hidden state computation of the large model and introduce a new attention layer to predict the next token. Sun et al. (2024a) employ the target model with a partial key-value cache as the draft model.

## B.2  MULTI-DRAFT SPECULATIVE DECODING

Many related works on Multi-Draft Speculative Decoding (MDSD) have been introduced in other sections. This paper focuses on the single-step Multi-Draft scenario. When MDSD generates multiple steps, with each step involving multiple drafts, it forms a tree structure. Sequoia (Chen et al., 2024) propose a dynamic programming algorithm to search for the optimal tree topology.

As the tree grows deeper, the acceptance probability of certain branches decreases. Cascade Speculative Drafting (Chen et al., 2023b) addresses this issue by assigning the largest draft model to generate draft tokens at shallower levels, which are more likely to be accepted, and gradually using smaller models to generate drafts for less relevant branches.

Khisti et al. (2024) studied the optimal acceptance rate for special case of sampling with replacement for $n = 2$ drafts, and obtained the following result:

$$\alpha^*(p, p_{\text{draft}}) = \min_{H \subset \Sigma} \left\{ \sum_{i \in H} p(i) + \left( \sum_{i \in \Sigma \setminus H} q(s) \right)^2 + 2 \left( \sum_{i \in H} q(s) \right) \left( \sum_{i \in \Sigma \setminus H} q(s) \right) \right\} . \quad (34)$$

This is essentially the same as our result (8) under this special case. However, our theory is more general, without any assumption on the draft sampling methods or the number of draft tokens.

## B.3  MULTI-STEP SPECULATIVE DECODING

The basic single-step, single-draft speculative decoding, as introduced in Section 2.1, can be applied to multiple steps, with each step having only one draft and an independent verification process (Leviathan et al., 2023; Chen et al., 2023a). However, such an approach of repeatedly applying single-step verification is not optimal for the multi-step scenario. Some works, such as Sun et al. (2024d); Hu & Huang (2024); Sun et al. (2024c), have designed better verification algorithms specifically for the multi-step setting. These algorithms are tailored for the multi-step scenario while remaining compatible with the single-step case, reducing to the basic speculative sampling algorithm when applied to a draft sequence of length 1.

Multi-step speculative decoding and multi-draft speculative decoding represent different directions for improvement.

As shown in Figure 3, Sun et al. (2024e); Khisti et al. (2024) and our work improve speculative decoding from the multi-draft perspective. When there is only a single draft, it reduces to the case in Leviathan et al. (2023); Chen et al. (2023a). On the other hand, Sun et al. (2024d); Hu & Huang

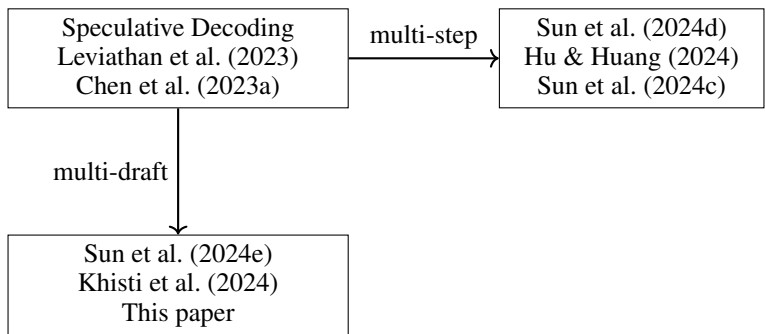

Figure 3: Different directions for improving speculative decoding.

(2024); Sun et al. (2024c) enhance speculative decoding from the multi-step perspective. When there is only a single step, it reduces to the case in Leviathan et al. (2023); Chen et al. (2023a).

Combining both improvements in the multi-draft and multi-step scenario would be ideal, and could be a direction for future research.

## C  PROOFS

*Proof of Lemma 1.* Let $f_1(C)$ and $f_2(S)$ denote the objective function values of (7) and (9), respectively. Let $v_1 = f_1(C^*)$ and $v_2 = f_2(S^*)$ be the optimal objective function values.

First, we show that the optimal solution of (7) is feasible for (9). Define $S_{i,\bar{i}} = C^*_{i,\bar{i}}$ for $\bar{i} \in A_i$ and $S_{i,\bar{i}} = 0$ for $\bar{i} \notin A_i$. This solution maintains the objective function value, i.e., $f_2(S) = f_1(C^*)$. Therefore, $v_2 = f_2(S^*) \geq f_2(S) = f_1(C^*) = v_1$.

Next, we show that the optimal solution of (9) is feasible for (7). Define $p^{\mathrm{res}}(i) = p(i) - \sum_{\bar{i} \in \Sigma^n} S^*_{i,\bar{i}} \geq 0$ for $i \in \Sigma$ and $p^{\mathrm{res}}_{\mathrm{draft}}(\bar{i}) = p_{\mathrm{draft}}(\bar{i}) - \sum_{i \in \Sigma} S^*_{i,\bar{i}} \geq 0$ for $\bar{i} \in \Sigma^n$. We have $\sum_{\bar{i} \in \Sigma^n} p^{\mathrm{res}}_{\mathrm{draft}}(\bar{i}) = \sum_{i \in \Sigma} p^{\mathrm{res}}(i)$. Define $C_{i,\bar{i}} = S^*_{i,\bar{i}} + \frac{p^{\mathrm{res}}(i) p^{\mathrm{res}}_{\mathrm{draft}}(\bar{i})}{\sum_{i \in \Sigma} p^{\mathrm{res}}(i)} \geq S^*_{i,\bar{i}}$. This solution has a larger objective function value, i.e., $f_1(C) \geq f_1(S^*)$. Therefore, $v_1 = f_1(C^*) \geq f_1(C) \geq f_2(S^*) = v_2$.

Combining the two parts, we have $v_1 = v_2$, which proves the equivalence of the two formulations. □

*Proof of Theorem 3.* For $\bar{i} = (\bar{i}_1, \ldots, \bar{i}_n) \in \Sigma^n$, we have $p_{\mathrm{draft}}(\bar{i}) = \prod_{j=1}^{n} q(\bar{i}_j)$. The function $Q(H) = \sum_{\bar{i} \in H^n} p_{\mathrm{draft}}(\bar{i})$ represents the probability that all $n$ samples drawn with replacement are in the set $H$. Therefore, $Q(H) = (\sum_{x \in H} q(x))^n$.

Consider the convex function $g(x) = x^n$. To prove the $q$-convexity of $Q$, it suffices to show that for all $H \subset \Sigma$ and $x, y \in \Sigma \setminus H$ with $x \neq y$, we have:

$$\frac{(Q(H) + q(x))^n - Q(H)^n}{q(x)} \leq \frac{(Q(H) + q(x) + q(y))^n - (Q(H) + q(x))^n}{q(y)} \tag{35}$$

This can be rewritten as:

$$\frac{g(Q(H) + q(x)) - g(Q(H))}{q(x)} \leq \frac{g(Q(H) + q(x) + q(y)) - g(Q(H) + q(x))}{q(y)} \tag{36}$$

Note that both sides are finite differences of the convex function $g$. Define $a = Q(x)$, $b = Q(x) + q(x)$, and $c = Q(x) + q(x) + q(y)$. It suffices to show that:

$$\frac{g(b) - g(a)}{b - a} \leq \frac{g(c) - g(b)}{c - b} \tag{37}$$

This follows directly from the convexity of $g$. □

*Proof of Theorem 4.* The function $Q(H) = \sum_{\vec{i} \in H^n} p_{\text{draft}}(\vec{i})$ represents the probability that all $n$ samples drawn without replacement are in the set $H$. To handle the more complex case of sampling without replacement, we use generating functions.

Define the generating function $G_H(t) = \prod_{i \in H}(1 + q(i)t)$ and the coefficient $W_{n,H} = \text{Coeff}_{t^n} G_H(t)$. Note that $Q(H) = \frac{W_{n,H}}{W_{n,\Sigma}}$.

The coefficients satisfy the following recurrence relation:

$$W_{n,H\cup\{x\}} = \text{Coeff}_{t^n} G_{H\cup\{x\}}(t) \tag{38}$$
$$= \text{Coeff}_{t^n} G_H(t) + q(x)tG_H(t) \tag{39}$$
$$= W_{n,H} + q(x)W_{n-1,H} \tag{40}$$

To prove the $q$-convexity of $Q$, it suffices to show that for all $H \subset \Sigma$ and $x, y \in \Sigma \setminus H$ with $x \neq y$, we have:

$$\frac{W_{n,H\cup\{x\}} - W_{n,H}}{q(x)W_{n,\Sigma}} \leq \frac{W_{n,H\cup\{x,y\}} - W_{n,H\cup\{x\}}}{q(y)W_{n,\Sigma}} \tag{41}$$

Applying the recurrence relation, it suffices to show that:

$$W_{n-1,H} \leq W_{n-1,H\cup\{x\}} \tag{42}$$

Applying the recurrence relation again, it suffices to show that:

$$0 \leq q(x)W_{n-2,H} \tag{43}$$

This holds because the coefficients of $G$ are always non-negative, i.e., $W_{n-2,H} \geq 0$. $\square$

*Proof of Theorem 5.* It suffices to show that for all $H \subset \Sigma$ and $x, y \in \Sigma \setminus H$ with $x \neq y$, we have:

$$Q(x|H) \leq Q(x|H \cup \{y\}) \tag{44}$$

By the $q$-convexity of $Q$, we have:

$$\frac{Q(x|H)}{q(x)} \leq \frac{Q(y|H \cup \{x\})}{q(y)} \tag{45}$$

Therefore,

$$\frac{Q(x|H)}{q(x)} \leq \frac{Q(x|H) + Q(y|H \cup \{x\})}{q(x) + q(y)} = \frac{Q(H \cup \{x,y\}) - Q(H)}{q(x) + q(y)} \leq \frac{Q(y|H \cup \{x\})}{q(y)} \tag{46}$$

Similarly, by symmetry, we can reverse $x$ and $y$ to obtain:

$$\frac{Q(y|H)}{q(y)} \leq \frac{Q(H \cup \{x,y\}) - Q(H)}{q(x) + q(y)} \leq \frac{Q(x|H \cup \{y\})}{q(x)} \tag{47}$$

Therefore,

$$\frac{Q(x|H)}{q(x)} \leq \frac{Q(H \cup \{x,y\}) - Q(H)}{q(x) + q(y)} \leq \frac{Q(x|H \cup \{y\})}{q(x)} \tag{48}$$

This implies that:

$$Q(x|H) \leq Q(x|H \cup \{y\}) \tag{49}$$

$\square$

*Proof of Theorem 6.* We have:

$$f(x|H) = p(x) - Q(x|H) \tag{50}$$
$$= p(x)(1 - \frac{q(x)}{p(x)}\frac{Q(x|H)}{q(x)}) \leq 0 \tag{51}$$

Therefore,

$$\frac{f(x|H)}{p(x)} = 1 - \frac{q(x)}{p(x)}\frac{Q(x|H)}{q(x)} \leq 0 \tag{52}$$

By assumption, $\frac{q(x)}{p(x)} \leq \frac{q(y)}{p(y)}$. By the $q$-convexity of $Q$, we have $\frac{Q(x|H)}{q(x)} \leq \frac{Q(y|H\cup\{x\})}{q(y)}$. Therefore,

$$\frac{q(y)}{p(y)}\frac{Q(y|H\cup\{x\})}{q(y)} \geq \frac{q(x)}{p(x)}\frac{Q(x|H)}{q(x)} \tag{53}$$

It follows that:

$$\frac{f(y|H\cup\{x\})}{p(y)} = 1 - \frac{q(y)}{p(y)}\frac{Q(y|H\cup\{x\})}{q(y)} \tag{54}$$

$$\leq \frac{f(x|H)}{p(x)} \leq 0 \tag{55}$$

$$\square$$

*Proof of Theorem 7.* We first prove that the acceptance rate of the greedy method is:

$$\alpha^{\text{Greedy}}(p, p_{\text{draft}}) \tag{56}$$

$$= \sum_{i\in\Sigma}\sum_{\bar{i}\in A_i} \pi^{\text{Greedy}}_{p,p_{\text{draft}}}(i,\bar{i}) \tag{57}$$

$$= \sum_{i\in\Sigma}\sum_{\bar{i}\in A_i} \pi^{\text{Greedy}}_{p,p_{\text{draft}}}(i|\bar{i})p_{\text{draft}}(\bar{i}) \tag{58}$$

$$= \sum_{i\in\text{Top}_{n-1}(q)}\sum_{\bar{i}\in A_i} \pi^{\text{Greedy}}_{p,p_{\text{draft}}}(i|\bar{i})p_{\text{draft}}(\bar{i}) \tag{59}$$

$$+ \sum_{i\in\Sigma\backslash\text{Top}_{n-1}(q)}\sum_{\bar{i}\in A_i} \pi^{\text{Greedy}}_{p,p_{\text{draft}}}(i|\bar{i})p_{\text{draft}}(\bar{i}) \tag{60}$$

$$= \sum_{i\in\text{Top}_{n-1}(q)}\sum_{\bar{i}_n\in\Sigma} \pi^{*}_{p,q^{\neg\text{Top}_{n-1}(q)}}(i|\bar{i}_n)q^{\neg\text{Top}_{n-1}(q)}(\bar{i}_n) \tag{61}$$

$$+ \sum_{i\in\Sigma\backslash\text{Top}_{n-1}(q)} \pi^{\text{Greedy}}_{p,p_{\text{draft}}}(i|(\text{Top}_{n-1}(q),i))q^{\neg\text{Top}_{n-1}(q)}(i) \tag{62}$$

$$= \sum_{i\in\text{Top}_{n-1}(q)} p(i) \tag{63}$$

$$+ \sum_{i\in\Sigma\backslash\text{Top}_{n-1}(q)} \pi^{*}_{p,q^{\neg\text{Top}_{n-1}(q)}}(i|i)q^{\neg\text{Top}_{n-1}(q)}(i) \tag{64}$$

$$= \sum_{i\in\text{Top}_{n-1}(q)} p(i) + \sum_{i\in\Sigma\backslash\text{Top}_{n-1}(q)} \min(p(i), q^{\neg\text{Top}_{n-1}(q)}(i)) \tag{65}$$

Note that $\sum_{i\in\text{Top}_{n-1}(q)} \min(p(i), q^{\neg\text{Top}_{n-1}(q)}(i)) = \sum_{i\in\text{Top}_{n-1}(q)} \min(p(i), 0) = 0$.

Next, we compute the optimal acceptance rate. Note that when $\text{Top}_{n-1}(q) \not\subseteq H$, we must have $Q(H) = 0$. When $\text{Top}_{n-1}(q) \subseteq H$, we have $Q(H) = \sum_{i\in H} q^{\neg\text{Top}_{n-1}(q)}(i)$. Therefore,

$$\alpha^{*}(p, p_{\text{draft}}) \tag{66}$$

$$= 1 + \min_{H\subset\Sigma} P(H) - Q(H) \tag{67}$$

$$= 1 + \min_{H\subset\Sigma, \text{s.t. } \text{Top}_{n-1}(q)\subseteq H} \sum_{i\in H} p(i) - q^{\neg\text{Top}_{n-1}(q)}(i) \tag{68}$$

The optimal set is $H^{*} = \{i\in\Sigma | q^{\neg\text{Top}_{n-1}(q)}(i) \geq p(i)\} \cup \text{Top}_{n-1}(q)$. In this case,

$$\alpha^{*}(p, p_{\text{draft}}) \tag{69}$$

$$=1 - \sum_{i \in \Sigma}(q^{\neg\,\mathrm{Top}_{n-1}(q)}(i) - p(i))_+ + \sum_{i \in \mathrm{Top}_{n-1}(q)} p(i) \tag{70}$$

$$= \sum_{i \in \Sigma} \min(p(i), q^{\neg\,\mathrm{Top}_{n-1}(q)}(i))_+ + \sum_{i \in \mathrm{Top}_{n-1}(q)} p(i) \tag{71}$$

$$\square$$

### C.1 DERIVATION OF THE DUAL PROBLEM

We start from the primal problem (9):

$$
\begin{aligned}
\max_{S \in \mathbb{R}^{\Sigma \times \Sigma^n}} &\sum_{i \in \Sigma} \sum_{\bar{i} \in \Sigma^n} S_{i,\bar{i}} \\
\text{s.t.} \quad &\sum_{\bar{i} \in \Sigma^n} S_{i,\bar{i}} \leq p(i) && \forall i \in \Sigma \\
&\sum_{i \in \Sigma} S_{i,\bar{i}} \leq p_{\mathrm{draft}}(\bar{i}) && \forall \bar{i} \in \Sigma^n \\
&S_{i,\bar{i}} \geq 0 && \forall i \in \Sigma, \bar{i} \in \Sigma^n \\
&S_{i,\bar{i}} = 0 && \forall i \in \Sigma, \bar{i} \notin A_i
\end{aligned} \tag{72}
$$

We introduce dual variables $y_i$ for each constraint $\sum_{\bar{i} \in \Sigma^n} S_{i,\bar{i}} \leq p(i)$ and $z_{\bar{i}}$ for each constraint $\sum_{i \in \Sigma} S_{i,\bar{i}} \leq p_{\mathrm{draft}}(\bar{i})$. The Lagrangian function is:

$$L(S, y, z) = \sum_{i \in \Sigma} \sum_{\bar{i} \in \Sigma^n} S_{i,\bar{i}} \tag{73}$$

$$+ \sum_{i \in \Sigma} y_i\left(p(i) - \sum_{\bar{i} \in \Sigma^n} S_{i,\bar{i}}\right) \tag{74}$$

$$+ \sum_{\bar{i} \in \Sigma^n} z_{\bar{i}}\left(p_{\mathrm{draft}}(\bar{i}) - \sum_{i \in \Sigma} S_{i,\bar{i}}\right) \tag{75}$$

The dual function is:

$$
\begin{aligned}
g(y, z) = \max_{S \in \mathbb{R}^{\Sigma \times \Sigma^n}} &\; L(S, y, z) \\
\text{s.t.} \quad &S_{i,\bar{i}} \geq 0 \quad \forall i \in \Sigma, \bar{i} \in \Sigma^n \\
&S_{i,\bar{i}} = 0 \quad \forall i \in \Sigma, \bar{i} \notin A_i
\end{aligned} \tag{76}
$$

Rearranging the Lagrangian function:

$$L(S, y, z) = \sum_{i \in \Sigma} \sum_{\bar{i} \in \Sigma^n}(1 - y_i - z_{\bar{i}})S_{i,\bar{i}} \tag{77}$$

$$+ \sum_{i \in \Sigma} y_i p(i) + \sum_{\bar{i} \in \Sigma^n} z_{\bar{i}} p_{\mathrm{draft}}(\bar{i}) \tag{78}$$

For the dual function to be bounded, we must have:

$$1 - y_i - z_{\bar{i}} \leq 0, \forall i \in \Sigma, \bar{i} \in A_i \tag{79}$$

Therefore, the dual problem is:

$$
\begin{aligned}
\min_{y \in \mathbb{R}^{\Sigma}, z \in \mathbb{R}^{\Sigma^n}} &\sum_{i \in \Sigma} y_i p(i) + \sum_{\bar{i} \in \Sigma^n} z_{\bar{i}} p_{\mathrm{draft}}(\bar{i}) \\
\text{s.t.} \quad &y_i + z_{\bar{i}} \geq 1 && \forall i \in \Sigma, \bar{i} \in A_i \\
&y_i \geq 0 && \forall i \in \Sigma \\
&z_{\bar{i}} \geq 0 && \forall \bar{i} \in \Sigma^n
\end{aligned} \tag{80}
$$

This completes the derivation of the dual problem.

# D    ADDITIONAL ILLUSTRATION

Illustration of single-step draft tokens generation and verification:

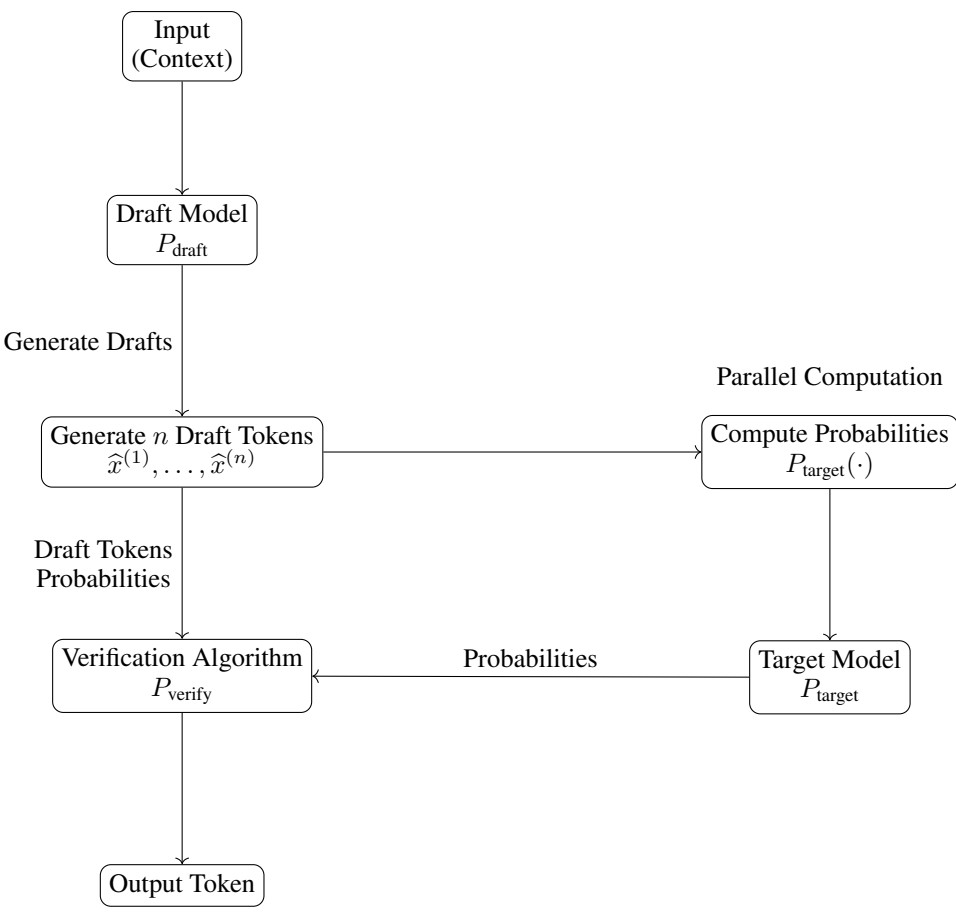

Pseudo code for apply multi-draft speculative sampling for multiple steps, with arbitrary tree topology.

```python
def multi_draft_speculative_decoding(prompt,
                                     tree_topology,
                                     draft_model,
                                     target_model):
    """
    Multi-Draft Speculative Decoding algorithm for accelerating
    language model inference.

    Example tree_topology:
        tree_topology = [
            [0], [1], [2], [3],  # First level: 4 branches
            [0,0], [0,1], [0,2], [1,0], [1,1],
            [2,0], [2,1], [3,0],  # Second level
            [0,0,0], [0,0,1], [0,0,2], [0,1,0],
            [0,1,1], [0,2,0], [0,2,1], [1,0,0], # Third level
            [0,0,0,0], [0,0,0,1], [0,0,0,2],  # Fourth level
            [0,0,0,0,0], [0,0,0,0,1]  # Fifth level
        ]
    This is the default EAGLE tree structure where each number
    represents which draft token to use at each level.
    """
```

```python
        # Initialize dictionaries to store drafts and distributions
        drafts = {}
        draft_distributions = {}
        target_distributions = {}

        # Generate drafts for each prefix in the tree topology
        for prefix in [[]] + tree_topology:
            children = get_children(tree_topology, prefix)
            if not children:
                continue  # Skip if no expandable children paths
            # Generate drafts and corresponding distributions
            # e.g. sampling with/without replacement or Section 5.1
            (
                drafts[tuple(prefix)],
                draft_distributions[tuple(prefix)]
            ) = generate_draft_tokens(draft_model, prefix)

        # Compute probability distributions from target model for all drafts
        target_distributions = compute_target_distributions(target_model, drafts)

        # Start verification and generation process
        prefix = []
        while True:
            if tuple(prefix) not in drafts:
                break  # End if no available drafts
            # e.g. RRS or K-Seq or Section 5.2
            output_token = verification(
                drafts[tuple(prefix)],
                draft_distributions[tuple(prefix)],
                target_distributions[tuple(prefix)]
            )
            prefix.append(output_token)

        # Return the generated sequence
        return prefix

def get_children(tree_topology, prefix):
    """
    Get all child paths in tree_topology that extend the given
    prefix by one token.

    Examples:
        tree_topology = [[0], [1], [0,0], [0,1], [1,0]]
        get_children([], tree_topology) -> [[0], [1]]
        get_children([0], tree_topology) -> [[0,0], [0,1]]
        get_children([1], tree_topology) -> [[1,0]]
        get_children([0,0], tree_topology) -> []
    """
    return [
        path for path in tree_topology
        if len(path) == len(prefix) + 1 and path[:len(prefix)] == prefix
    ]
```

Unfolded definition of verify algorithm for greedy draft construction.

$$
\begin{aligned}
&\pi_{p,p_{\text{draft}}}^{\text{Greedy}}(i|\bar{i}) \\
&=\pi_{p,q^{\neg\text{Top}_{n-1}(q)}}^{*}(i|\bar{i}_n) \\
&=\begin{cases}
\min(\dfrac{p(i)(1-\sum_{j\in\text{Top}_{n-1}(q)}q(j))}{q(i)},1) & i=\bar{i}_n \\
(1-\dfrac{p(\bar{i}_n)(1-\sum_{j\in\text{Top}_{n-1}(q)}q(j))}{q(\bar{i}_n)})+\dfrac{p(i)(1-\sum_{j\in\text{Top}_{n-1}(q)}q(j))}{\sum_{z\in\Sigma}(p(z)(1-\sum_{j\in\text{Top}_{n-1}(q)}q(j))-\mathbb{I}(z\notin\text{Top}_{n-1}(q))q(z))_+} & i\in\text{Top}_{n-1}(q) \\
(1-\dfrac{p(\bar{i}_n)(1-\sum_{j\in\text{Top}_{n-1}(q)}q(j))}{q(\bar{i}_n)})+\dfrac{(p(i)(1-\sum_{j\in\text{Top}_{n-1}(q)}q(j))-q(i))_+}{\sum_{z\in\Sigma}(p(z)(1-\sum_{j\in\text{Top}_{n-1}(q)}q(j))-\mathbb{I}(z\notin\text{Top}_{n-1}(q))q(z))_+} & i\neq\bar{i}_n, i\notin\text{Top}_{n-1}(q)
\end{cases}
\end{aligned}
\tag{81}
$$

## E  SUMMARY OF NOTATIONS

- $\Sigma$: The vocabulary set
- $\Delta_\Sigma$: The probability simplex over vocabulary $\Sigma$
- $[n]$: The set 1,...,n
- $P_{\text{target}}(\cdot|x_1,x_2,...,x_m)$: The target model, a probabilistic model that predicts the probability of the next word given the context
- $P_{\text{draft}}(\cdot|x_1,x_2,...,x_m)$: The draft model used to generate candidate tokens
- $P_{\text{verify}}(\cdot|\hat{x}^{(1)},\ldots\hat{x}^{(n)})$: The verification algorithm that selects the final output token from the draft tokens
- $p(\cdot)=P_{\text{target}}(\cdot|x_1,x_2,...,x_m)$: Shorthand for the target distribution
- $p_{\text{draft}}$: The distribution of draft tokens
- $\pi\in\Pi(p,p_{\text{draft}})$: A joint distribution with marginal distributions $p$ and $p_{\text{draft}}$
- $\pi_{p,p_{\text{draft}}}^{*}\in\Pi(p,p_{\text{draft}})$: The optimal transport joint distribution
- $\alpha^{*}(p,p_{\text{draft}})$: The optimal acceptance rate
- $A_i:=\{\bar{i}\in\Sigma^n|\exists j\in[n],\bar{i}_j=i\}$: The incidence set for token $i$
- $q(\cdot)$: The shorthand notation of the output distribution of the draft model
- $q^{\neg\bar{i}_1,...,\bar{i}_{j-1}}(x)$: The probability of token $x$ when sampling without replacement, excluding previously selected tokens
- $\text{Res}^{p-q}\in\Delta_\Sigma$: The residual distribution
- $\pi_{p,p_{\text{draft}}}^{\text{RRS,w}},\pi_{p,p_{\text{draft}}}^{\text{RRS,wo}}$: The RRS verification algorithms for with/without replacement sampling
- $\alpha^{\text{RRS,w}}(p,p_{\text{draft}}),\alpha^{\text{RRS,wo}}(p,p_{\text{draft}})$: Acceptance rates for RRS with/without replacement
- $\beta_{p,q}(\rho)$: A function used in the K-SEQ algorithm
- $\pi_{p,p_{\text{draft}}}^{\text{K-SEQ}}$: The K-SEQ verification algorithm
- $\alpha^{\text{K-SEQ}}(p,p_{\text{draft}})$: Acceptance rate for the K-SEQ algorithm
- $P(H)=\sum_{i\in H}p(i)$: Sum of target probabilities over set $H$
- $Q(H)=\sum_{\bar{i}\in H^n}p_{\text{draft}}(\bar{i})$: Sum of draft probabilities over set $H^n$
- $f(H)=P(H)-Q(H)$: Difference between target and draft probabilities over set $H$
- $\pi_{p,p_{\text{draft}}}^{\text{Greedy}}$: The greedy verification algorithm
- $\alpha^{\text{Greedy}}(p,p_{\text{draft}})$: Acceptance rate for the greedy draft sampling method
- $C_{i,j}$: Matrix representation of joint distribution
- $(p(i)-p_{\text{draft}}(i))_+$: The positive part of the difference
- $G_H(t)$: Generating function defined as $\prod_{i\in H}(1+q(i)t)$
- $W_{n,H}$: Coefficient of $t^n$ in $G_H(t)$

- $f(x|H)$: The marginal value of element x with respect to set $H$
- $\text{Top}_{n-1}(q)$: The top n-1 tokens according to probability in $q$
- $S \in \mathbb{R}^{\Sigma \times \Sigma^n}$: Variables in the equivalent LP formulation
- $y \in \mathbb{R}^{\Sigma}, z \in \mathbb{R}^{\Sigma^n}$: Dual variables
- $\pi(\cdot|j)$: The conditional distribution given $j$
- $\pi_{i,j}$: Individual elements of the joint distribution matrix
- $\bar{i}_{2:}$: The sequence $\bar{i}$ with the first element removed
- $H_i$: Sets constructed by adding elements one by one
- $\sigma$: An ordering of $\Sigma$ used in the efficient computation algorithm
- $\text{Coeff}_{t^n}$: Coefficient of $t^n$ in a generating function

# F  ADDITIONAL EXPERIMENTS

Table 3: Average generation length $\tau$ of different MDSD methods and their ratio $\Delta$ on MT-Bench based on Eagle framework.

| Method | # Drafts = 2, # Steps = 4 | | # Drafts = 4, # Steps = 3 | | EAGLE default sparse tree | |
|---|---|---|---|---|---|---|
| | $\tau$ | $\Delta$ | $\tau$ | $\Delta$ | $\tau$ | $\Delta$ |
| $T = 0.1$ | | | | | | |
| RRS w/ replacement | $3.04 \pm 0.02$ | - | $2.83 \pm 0.02$ | - | $3.19 \pm 0.03$ | - |
| RRS w/o replacement | $3.27 \pm 0.02$ | $1.07\,(\pm 0.02)\times$ | $2.96 \pm 0.02$ | $1.05\,(\pm 0.01)\times$ | $3.42 \pm 0.03$ | $1.07\,(\pm 0.02)\times$ |
| SpecHub | $3.63 \pm 0.02$ | $1.19\,(\pm 0.02)\times$ | - | - | - | - |
| Greedy | $3.62 \pm 0.02$ | $1.19\,(\pm 0.02)\times$ | $3.39 \pm 0.01$ | $1.20\,(\pm 0.02)\times$ | $3.70 \pm 0.03$ | $1.16\,(\pm 0.02)\times$ |
| $T = 0.6$ | | | | | | |
| RRS w/ replacement | $3.22 \pm 0.02$ | - | $3.11 \pm 0.01$ | - | $3.41 \pm 0.02$ | - |
| RRS w/o replacement | $3.52 \pm 0.02$ | $1.09\,(\pm 0.02)\times$ | $3.39 \pm 0.01$ | $1.09\,(\pm 0.01)\times$ | $3.71 \pm 0.02$ | $1.09\,(\pm 0.02)\times$ |
| SpecHub | $3.52 \pm 0.02$ | $1.09\,(\pm 0.02)\times$ | - | - | - | - |
| Greedy | $3.52 \pm 0.02$ | $1.09\,(\pm 0.02)\times$ | $3.45 \pm 0.01$ | $1.11\,(\pm 0.01)\times$ | $3.66 \pm 0.02$ | $1.07\,(\pm 0.02)\times$ |
| $T = 1.0$ | | | | | | |
| RRS w/ replacement | $3.14 \pm 0.02$ | - | $3.09 \pm 0.01$ | - | $3.22 \pm 0.02$ | - |
| RRS w/o replacement | $3.22 \pm 0.02$ | $1.02\,(\pm 0.02)\times$ | $3.25 \pm 0.01$ | $1.05\,(\pm 0.01)\times$ | $3.43 \pm 0.02$ | $1.06\,(\pm 0.02)\times$ |
| SpecHub | $3.35 \pm 0.02$ | $1.07\,(\pm 0.02)\times$ | - | - | - | - |
| Greedy | $3.33 \pm 0.02$ | $1.06\,(\pm 0.02)\times$ | $3.34 \pm 0.01$ | $1.08\,(\pm 0.01)\times$ | $3.33 \pm 0.02$ | $1.03\,(\pm 0.02)\times$ |

**Remark 8.** *For # Drafts = 2, # Steps = 4, and $T = 0.6$, three methods - RRS without replacement, SpecHub, and Greedy - show similar average generation lengths. After truncating to two decimal places, they appear to be the same. However, they are actually different numbers: 3.51781, 3.51944, 3.51975.*

