# OpenReview forum: "Towards Optimal Multi-draft Speculative Decoding"
_ICLR.cc/2025/Conference — ICLR 2025 Poster_

### Official Review · Reviewer_DHWn · 2024-10-21

**Soundness:** 3
**Presentation:** 2
**Contribution:** 2
**Rating:** 5
**Confidence:** 3

**Summary:**

This paper studies the problem of multi-draft speculative decoding (MDSD), where the draft model provides multiple draft tokens for each draft position. The authors first provide a way to compute the optimal acceptance rate. Then, they measure the theoretical upper bound of MDSD with large vocab size and quantify the gap between existing verification algorithms and this bound. Besides, the authors also provide a greedy draft sampling methods to approach the theoretical upper bound of MDSD.

**Strengths:**

1. The idea of transforming the problem into a subset selection problem and considering the dual of the problem is novel and makes sense.
2. The authors rigorously give some theoretical findings, including the upper bound of MDSD and a efficient method to compute the theoretical acceptance rate.
3. The authors propose a greedy draft sampling method and conduct extensive experiments to demonstrate its effectiveness.

**Weaknesses:**

While this paper provides rigorous theory and analysis, I think there exists some weakness to further improve the manuscript.

1. The motivation and background should be clearly illustrated, and the authors could consider add some intuitive examples and figures to improve the presentation.
2. Please discuss the connections to the related works. It is confusing for readers without the knowledge about SpecTr. Besides,  please give a clear notation section. For example, the number of draft tokens for each draft position and the number of draft length should be clarified.
3. Please describe the whole algorithm of the greedy draft sampling method. For example, after constructing the first  $n$ draft tokens for the first draft position, how we can construct the following draft tokens? Besides, this algorithm is similar to the top-k sampling, with only an additional random sampled token. The authors should discuss their difference.
4. The experiments could be strengthen by evaluating the block-wise mean accepted tokens and real-world speedup. Besides, more experiments with different model scales (e.g. 33B, 70B) and different benchmarks (e.g. MT-Bench [1] and Spec-Bench[2]) are necessary to demonstrate the conclusions.

[1] Zheng, Lianmin, et al. "Judging llm-as-a-judge with mt-bench and chatbot arena." Advances in Neural Information Processing Systems 36 (2023): 46595-46623.

[2] Xia, Heming, et al. "Unlocking efficiency in large language model inference: A comprehensive survey of speculative decoding." arXiv preprint arXiv:2401.07851 (2024).

**Questions:**

1. In the real-world applications of speculative decoding, the acceptance rate of different position is usually not i.i.d. I wonder if this will affects the proposed theory and greedy draft sampling methods.
2. In Table 1, some results of empirical $\alpha$ is even higher than the theoretical upper bound. Could you please provide a detailed explanation?
3. In ablation study 1, the authors show an interesting phenomenon that the impact of temperature is non-monotonic. Different methods consistently show a turn around temperature $T=0.9$. Could you please provide a detailed explanation?
4. Can proposed greedy draft sampling methods adapt to other retrieval-based speculative decoding methods? (e.g. Lookahead Decoding [1] and REST [2])

[1] Fu, Yichao, et al. "Break the sequential dependency of llm inference using lookahead decoding." arXiv preprint arXiv:2402.02057 (2024).

[2] He, Zhenyu, et al. "Rest: Retrieval-based speculative decoding." arXiv preprint arXiv:2311.08252 (2023).

---

> ### Author Response · Authors · 2024-11-15
> **Author's Response (1)**
>
> Thank you for volunteering your time and effort to review our paper. We have carefully considered your feedback and improve the presentation of the manuscript. Below, we address your comments point by point.
>
> > The motivation and background should be clearly illustrated, and the authors could consider add some intuitive examples and figures to improve the presentation.
>
> We have added additional illustrations to enhance the presentation and help readers better understand our work.
>
> > Please discuss the connections to the related works. It is confusing for readers without the knowledge about SpecTr.
>
> We have included discussions on the connections to important related works in Appendix A and comparisons to other related works in Appendix B. If there are specific works you would like us to elaborate on regarding the connections, please let us know and we would be happy to include additional discussions.
>
> > Besides, please give a clear notation section. For example, the number of draft tokens for each draft position and the number of draft length should be clarified.
>
> We have added a notation section in Section E to clarify the symbols used in the paper. In the main text, $n$ represents the number of draft tokens for each draft position. The number of draft length is not explicitly discussed in our theory as our theory and algorithm works independently for each position. The specific values used in the experiments are described in the header of Table 2. For example, in the first column, the number of draft tokens for each draft position is 2 and the number of draft length is 4. In the second column, the number of draft tokens for each draft position is 4 and the number of draft length is 3. The third column corresponds to the more complex setting proposed in EAGLE, where the number of draft tokens and draft length vary for different branches. We hope these clarifications help improve the understanding of our notations.
>
> > Please describe the whole algorithm of the greedy draft sampling method. For example, after constructing the first draft tokens for the first draft position, how we can construct the following draft tokens?
>
> We provide pseudocode in Section D to help illustrate the algorithm. After constructing the first draft token, we append it to the prompt, feed it into the draft model, and obtain the next set of draft tokens through sampling. The process becomes more complex when considering a general tree topology where different branches may have different depths. In each step, the construction of draft tokens follows Section 5.1 and the verification of draft tokens follows Section 5.2. The specific implementation details, including parallelization for acceleration, can be found in the provided code.
>
> > Besides, this algorithm is similar to the top-k sampling, with only an additional random sampled token. The authors should discuss their difference.
>
> Our greedy draft construction method differs significantly from top-k sampling. Top-k sampling constrains the output space to k tokens and generates a single token, while our method does not constrain the output space and generates k tokens.
>
> > The experiments could be strengthen by evaluating the block-wise mean accepted tokens and real-world speedup.
>
> The real-world speedup results are already provided in the "Speed" column of Table 2. We have added the block-wise mean accepted tokens results in Section F, with the original data already available in the supplementary materials to ensure reproducibility.
>
> > Besides, more experiments with different model scales (e.g. 33B, 70B) and different benchmarks (e.g. MT-Bench [1] and Spec-Bench[2]) are necessary to demonstrate the conclusions.
>
> We have conducted experiments on 4 datasets and 4 model architectures, including the MT-Bench benchmark you mentioned (see Table 2). While we agree that more experiments with larger models and additional datasets would be beneficial, the computational cost quickly becomes prohibitive.
>
> We would appreciate it if you could kindly revisit your evaluation, taking into the consideration of MT-Bench, real-world speedup, and the newly added block-wise mean accepted tokens results in the paper.

---

> ### Author Response · Authors · 2024-11-15
> **Author's Response (2)**
>
> > In the real-world applications of speculative decoding, the acceptance rate of different position is usually not i.i.d. I wonder if this will affects the proposed theory and greedy draft sampling methods.
>
> Our work does not assume the acceptance rates at different positions to be i.i.d. at any point, so this does not affect our proposed theory or methods.
>
> > In Table 1, some results of empirical is even higher than the theoretical upper bound. Could you please provide a detailed explanation?
>
> The observed discrepancies are due to random errors and are not statistically significant. Speculative decoding inherently involves randomness, and whether a token is accepted is a random behavior. We have done our best to measure the statistical errors and faithfully report the results in each table and figure. In Table 1, although there are a few positive values, none of them are statistically significant at α=0.05. All statistically significant signals indicate that the empirical results are lower than the theoretical upper bound. We hope these explanations help clarify the results.
>
> > In ablation study 1, the authors show an interesting phenomenon that the impact of temperature is non-monotonic. Different methods consistently show a turn around temperature. Could you please provide a detailed explanation?
>
> We have experimentally observed the non-monotonic behavior of acceptance rate with respect to temperature in the speculative decoding process of current language models. Recently, independent researchers have discovered similar behaviors in"Temperature-Centric Investigation of Speculative Decoding with Knowledge Distillation, Figure 1, although their specific settings differ as they do not consider multi-draft scenarios. They found that the acceptance rate peaks at a temperature of 0.2, while in our Figure 1(a) and (c) with sampling with replacement, the acceptance rate is highest at temperatures between 0.1 and 0.3. We also found that this non-monotonicity depends on the dataset, as evident in Figure 1(b) where the behavior changes with a different dataset. Our work, with dense sampling in the 0.8-1.0 range, discovered new non-monotonic behaviors that were not observed in the "Temperature-Centric Investigation of Speculative Decoding with Knowledge Distillation" study.
>
> The non-monotonic phenomenon is a recently discovered phenomenon with dedicated and ongoing independent research still in its early stages. We do not yet know the exact cause of this behavior. Our findings, although not part of the 4 main contributions summarized in the introduction, provide additional evidence for this phenomenon.
>
> > Can proposed greedy draft sampling methods adapt to other retrieval-based speculative decoding methods? (e.g. Lookahead Decoding [1] and REST [2])
>
> Firstly, we note that Lookahead Decoding is not retrieval-based but rather attempts to consider multiple steps ahead. Our greedy draft method in this paper only considers the 1-step case when deriving the theoretical upper bound. Combining the strengths of both approaches is non-trivial and left for future work. Secondly, in principle, our method can be applied to REST: Retrieval-based speculative decoding. The authors designed the retrieval process to be deterministic, resulting in a delta distribution for the draft distribution. In this case, the optimal transport is trivial. It is possible to design non-delta draft distributions based on the retrieval results, in which case our greedy draft method can be applied. This can be explored in future work.
>
> We apologize for any misunderstandings and have addressed each comment individually.
>
> Once again, we appreciate your time and effort in reviewing our paper. We have incorporated your suggestions to improve the presentation of the paper.
>
> We would be immensely grateful if you could re-evaluate our paper considering the clarifications we have provided,
>
> Sincerely,
>
> The Authors

---

> > ### Comment · Reviewer_DHWn · 2024-11-19
> >
> > Thank you very much for your detailed explanation and prompt reply. I highly recognize the authors' efforts in developing the algorithm.
> >
> > However, the rebuttal does not fully address my concerns as follows:
> >
> > 1. (**clarity**) While I appreciate the authors' effort in improving the clarify of this paper, it is still difficult for readers to get the key idea, due to massive notations and equations. I strongly recommend the authors to focus on simplifying the notations and equations.
> > 2. (**contributions**) While the proposed methods and theoretical findings are rigorous and novel, the multi-draft framework remains trivial to me. In real-world applications, speculative decoding has been proven to be less effective with batch size > 1, even slow down the inference speed with batch size > 64. Therefore, the multi-draft framework, which brings additional overhead, may harm the overall throughput more severely in the high request settings.
> >
> > Other questions:
> > 1. For weakness 3, what I mean "top-k sampling" is using top k candidate tokens (k tokens in total) to construct multi-drafts. This vanilla method seems similar to the proposed method. Could the authors give a further explanation?
> > 2.  In section 4.2.2, the complexity of sorting is $O(\Sigma \log \Sigma)$. In the latest LLMs (e.g. Llama 3), the vocab size has been extended to 128k, which makes sorting more expensive. Could the authors give some empirical results of the time cost of the sorting operation? Maybe it is the bottleneck for the current algorithm.
> >
> > Based on these considerations, I will respectfully maintain my score. I sincerely hope your understanding.

---

> > > ### Author Response · Authors · 2024-11-19
> > > **Author's Response (1)**
> > >
> > > We sincerely appreciate your time and effort in reviewing our paper and engaging in this discussion. We would like to address your remaining concerns point by point.
> > >
> > > > While I appreciate the authors' effort in improving the clarify of this paper, it is still difficult for readers to get the key idea, due to massive notations and equations. I strongly recommend the authors to focus on simplifying the notations and equations.
> > >
> > > Thank you for acknowledging our existing efforts to improve the clarity of the paper. Regarding the notations and equations, they are inherent burdens that come from presenting rigorous and novel theoretical results. We believe that all the formulas are necessary to maintain the theoretical rigor, and we are not aware of any equations that can be removed without compromising the strictness. If you have specific suggestions on which parts we could remove while preserving the rigor, we would be more than happy to know.
> > >
> > > In addition, as per your request, we have already added a clear notation section. Please feel free to use it to navigate through the notations. We would be glad to know if there is anything else we could do to help you better get the key idea of the paper.
> > >
> > > > While the proposed methods and theoretical findings are rigorous and novel, the multi-draft framework remains trivial to me.
> > >
> > >
> > > We appreciate your recognition of the rigor and novelty of our methods and theoretical findings. The contribution of this paper, which simplifies an exponential complexity problem to $O(|\Sigma|\log|\Sigma|)$, is clearly a highly non-trivial result. As for the general field of multi-draft speculative sampling, it has helped many people and organizations in the real world to save costs and has made the capabilities of LLMs accessible to more people. The field continues to have new research works [1, 2, 3, 4, 5] that improve upon it, therefore we believe the field of multi-draft speculative sampling is also highly non-trivial.
> > >
> > > [1] Chen, Z., May, A., Svirschevski, R., Huang, Y., Ryabinin, M., Jia, Z., & Chen, B. (2024). Sequoia: Scalable, robust, and hardware-aware speculative decoding. NeurIPS 2024.
> > >
> > > [2] Sun, H., Chen, Z., Yang, X., Tian, Y., & Chen, B. (2024). Triforce: Lossless acceleration of long sequence generation with hierarchical speculative decoding. COLM, 2024.
> > >
> > > [3] Cai, T., Li, Y., Geng, Z., Peng, H., Lee, J. D., Chen, D., & Dao, T. (2024). Medusa: Simple llm inference acceleration framework with multiple decoding heads. ICML 2024.
> > >
> > > [4] Li, Y., Wei, F., Zhang, C., & Zhang, H. (2024). Eagle: Speculative sampling requires rethinking feature uncertainty. ICML 2024.
> > >
> > > [5] Li, Y., Wei, F., Zhang, C., & Zhang, H. (2024). Eagle-2: Faster inference of language models with dynamic draft trees. EMNLP 2024.
> > >
> > > > In real-world applications, speculative decoding has been proven to be less effective with batch size > 1, even slow down the inference speed with batch size > 64. Therefore, the multi-draft framework, which brings additional overhead, may harm the overall throughput more severely in the high request settings.
> > >
> > > To our understanding, the impact of batch size on speculative decoding is still an ongoing research topic with different viewpoints. For example, "MagicDec: Breaking the Latency-Throughput Tradeoff for Long Context Generation with Speculative Decoding" suggests that "an intelligent drafting strategy can achieve better speedup with increasing batch size" and "for moderate to large sequence lengths, speculative decoding can achieve all three objectives: increased throughput, reduced latency, and lossless accuracy." Regardless of the impact of batch size, it does not affect our study of the theoretical optimal acceptance rate and is orthogonal to our contributions.

---

> > > ### Author Response · Authors · 2024-11-19
> > > **Author's Response (2)**
> > >
> > > > For weakness 3, what I mean "top-k sampling" is using top k candidate tokens (k tokens in total) to construct multi-drafts. This vanilla method seems similar to the proposed method. Could the authors give a further explanation?
> > >
> > > Thank you for the clarification. We now understand that your "top-k sampling" refers to selecting the top k tokens with the highest logits from the draft model as draft tokens in speculative decoding. We would like to explain that this is not a popular method due to its low acceptance rate, even in the single draft setting. Note that even if the draft distribution and the target distribution are perfectly identical, using the tokens with the highest logits from the draft model as draft tokens may still fail to achieve an acceptance rate of 1. In contrast, using optimal transport can achieve an acceptance rate of 1. The same holds true in the multi-draft setting: even if the draft distribution and the target distribution are perfectly identical, using the top k tokens with the highest logits from the draft model as draft tokens may still fail to achieve an acceptance rate of 1. Due to this decline in acceptance rate, the method of selecting tokens with the highest logits is only commonly used when the temperature is 0, where optimal transport converges to this method.
> > >
> > > > In section 4.2.2, the complexity of sorting is $O(\Sigma \log \Sigma)$. In the latest LLMs (e.g. Llama 3), the vocab size has been extended to 128k, which makes sorting more expensive. Could the authors give some empirical results of the time cost of the sorting operation? Maybe it is the bottleneck for the current algorithm.
> > >
> > > In the models we tested, the maximum vocabulary size is about 150,000—larger than 128k—it takes only 7ms to sort using numpy. This is not a bottleneck for the system at all and is negligible compared to the language model. It is important to note that before our work, the complexity for the same problem is exponential, which is intractable. Our novel contribution makes it possible to compute the exact solution without approximation on such a scale for the first time.
> > >
> > > Once again, thank you for your time and effort in reviewing our paper and engaging in this discussion. If you could consider the clarifications we have provided, assess whether they address your remaining concerns, we would be deeply appreciative.
> > >
> > > Sincerely,
> > >
> > > Authors

---

> > > ### Author Response · Authors · 2024-11-23
> > > **Appreciating Your Feedback**
> > >
> > > We would appreciate your feedback on if any concerns remain as the discussion phase comes to a close. Thank you for your time and we are grateful for the discussion opportunity.
> > >
> > > Thanks!

---

### Official Review · Reviewer_WnL3 · 2024-11-04

**Soundness:** 3
**Presentation:** 3
**Contribution:** 2
**Rating:** 5
**Confidence:** 3

**Summary:**

This paper is concerned with acceptance rate of MDSD in LLMs. Authors derive the dual problem, and then prove it has an integer optimal solution, furthermore, they provide a greedy algorithm that, in some cases, perform better without replacement.

**Strengths:**

Paper is mathematically rigorous; it is relatively easy to follow and grasp new concepts. Authors do a good job of highlighting the drawbacks of previous work and offer solutions.

**Weaknesses:**

Some claims are optimistic, such as "the upper bound has never been computed before", I personally refrain from making such certain statements. Some contributions are minor, as an example, deriving the dual of an LP is not a contribution, yet it is claimed to be in the first bullet point of contributions. Although paper is mathematically mature, it borrows a lot from previous publications, in other words, novel theoretical contribution is minor.

**Questions:**

You stated that problem (equation 7) is intractable/difficult to solve. What algorithms did you use? modern LP algorithms such as interior point methods are quite capable at handling large problems (with exponential constraints), and recently there have been solvers implemented on GPU (see cuPDLD-C). Your greedy algorithm is another version of coin problem, in order for it to be optimal, the environment has to be canonical (see “Error Bounds and the Applicability of the Greedy Solution to the Coin-Changing Problem,”), I suggest you incorporate that in your proof. You have mentioned theoretical upper bound multiple times, however it is not explicitly defined, is it $\alpha^*$? I suggest you use Radix sort, which is linear in the size of input, and helps with the overall complexity of your problem.

---

> ### Author Response · Authors · 2024-11-15
> **Author's Response (1)**
>
> Thank you for volunteering your time and effort to review our paper. We apologize for any misunderstandings that have arisen. Based on your comments, we note that the complexity of the problem has not been fully conveyed. We will do our best to clarify the complexity to help you better understand our work. We will then address your other comments.
>
> ### Regarding the complexity of our work:
>
> > You stated that problem (equation 7) is intractable/difficult to solve. What algorithms did you use? modern LP algorithms such as interior point methods are quite capable at handling large problems (with exponential constraints), and recently there have been solvers implemented on GPU (see cuPDLD-C).
>
> The difficulty of solving the LP does not depend on the specific LP solver used, but rather on the scale of the problem. As stated in our paper, "The difficulty lies in the exponential number of variables and constraints."
>
> For example, let's conservatively assume the vocabulary size $|\Sigma| = 10^3$ (for reference, Llama's vocabulary size is already 32,000) and the number of multi-drafts $n = 3$. Then the variable $C$ in equation (7) has dimension $\mathbb{R}^{\Sigma\times\Sigma^n} = 10^{12}$. Just expressing the variable would require around 1 Terabyte of storage, which exceeds the memory of a typical CPU or GPU.
>
> The exponential growth in complexity prevents us from feasibly doing large scale experiments with the LP. This difficulty applies to any LP algorithm, including the interior point methods and cuPDLD-C that you mentioned, and they cannot resolve this issue. We hope this clarifies the complexity of the LP problem in equation (7).
>
> ### Addressing your other comments:
>
> > Some claims are optimistic, such as "the upper bound has never been computed before", I personally refrain from making such certain statements.
>
> We would like to clarify this point by quoting the original text: "For modern LLMs, where the vocabulary size is typically in the thousands, the optimal acceptance rate has never been computed." The preceding text was discussing RRS and K-SEQ, which are used for sampling with/without replacement.
>
> The intractability of the problem has been explained above. To the best of our knowledge, we have not seen any insights similar to our paper or any methods that reduce the exponential complexity to polynomial time, making it solvable.
>
> However, we acknowledge that there might be relevant works we have overlooked. If you are aware of any such prior works, we would greatly appreciate it if you could point them out. We are open to discussing and comparing our work with related literature.
>
> In the absence of identified prior works addressing this specific problem, we believe our claim of novelty stands. Nonetheless, we will be more cautious in our phrasing to avoid sounding overly certain.
>
> > Some contributions are minor, as an example, deriving the dual of an LP is not a contribution, yet it is claimed to be in the first bullet point of contributions.
>
> We would like to clarify this misunderstanding by quoting the original text of the first bullet point of contributions:
>
> "We transform the problem of solving the optimal acceptance rate corresponding to the optimal transport into a subset selection problem by considering the dual of the problem and then applying total unimodularity. This provides a novel perspective for understanding the efficiency of MDSD."
>
> According to the original text, our first contribution is "We transform the problem of solving the optimal acceptance rate corresponding to the optimal transport into a subset selection problem", and the method we use to make this contribution is "by considering the dual of the problem and then applying total unimodularity". The significance of our contribution is that "This provides a novel perspective for understanding the efficiency of MDSD."
>
> Therefore, deriving the dual is only a small step in proving our first contribution. Your previous comment only considered the step of deriving the dual, leading to the misunderstanding that "Some contributions are minor". By reading our original text, our conclusion is that our contributions are groundbreaking and important.
>
> We would appreciate it if you could kindly revisit your evaluation of the impact, taking the entire scope of this contribution into consideration.
>
> > Although paper is mathematically mature, it borrows a lot from previous publications, in other words, novel theoretical contribution is minor.
>
> We would like to clarify this misunderstanding by noting that citing many works is not a sufficient condition for minor contributions. While we indeed build upon many classic works, as evidenced by our extensive references, this is by no means a reason to consider our theoretical contributions as minor.
>
> All four of our contributions are novel and add significant new insights to the field of multi-draft speculative decoding.

---

> ### Author Response · Authors · 2024-11-15
> **Author's Response (2)**
>
> > Your greedy algorithm is another version of coin problem, in order for it to be optimal, the environment has to be canonical (see "Error Bounds and the Applicability of the Greedy Solution to the Coin-Changing Problem,"), I suggest you incorporate that in your proof.
>
> Firstly, our greedy draft token construction method in Section 5.1, which is a probability distribution, has nothing to do with the Coin-Changing Problem, which is an optimization problem.
>
> Additionally, the reviewer may be referring to $f(H) = P(H) − Q(H)$ in Section 4, but this algorithm is not called a greedy algorithm. It is an optimization problem, but it also has many differences than the Coin-Changing Problem.
>
> Definitions:
> - In Section 4: $\min_H \sum_{i\in H}p(i) − Q(H)$
> - Coin-Changing Problem: $\begin{aligned}\min\_{x\in\mathbb{Z}\_+^n} & \sum\_{i=1}^n c_i x_i \\\\ \text{s.t.}& \sum\_{i=1}^n a\_i x\_i = b\end{aligned}$
>
> Important differences:
> - The optimization functions are different: Q is a nonlinear function. There is no nonlinear function in the Coin-Changing Problem.
> - The constraints are different: the former has no constraints, while the Coin-Changing Problem has linear constraints.
> - The variable spaces are different: the Coin-Changing Problem is an integer programming problem, while our paper is a subset selection problem.
>
> We hope that the differences we have pointed out help resolve your misunderstanding.
>
> > You have mentioned theoretical upper bound multiple times, however it is not explicitly defined, is it $\alpha^\ast$?
>
> Yes, the theoretical acceptance rate upper bound is the optimal acceptance rate $\alpha^\ast$.
>
> > I suggest you use Radix sort, which is linear in the size of input, and helps with the overall complexity of your problem.
>
> We would like to clarify this misunderstanding by noting that radix sort runs in $O(kn)$ time, where k is the number of digits in each value of the input array and n is the size of the input array. This running time is worse than $O(n\log n)$ when the number of digits is larger, especially in the case of using floating-point numbers in experiments. Secondly, we are dealing with floating-point numbers, which, unlike integers, cannot be directly used with Radix in machine representation. Implementing radix sort correctly for floating-point numbers is non-trivial and introduces complexity that is unrelated to the core contributions of our paper. Finally, the largest vocabulary size of the models we tested is 150,000, and it only takes 7ms to sort using numpy, which is not a system bottleneck and is very small compared to the language model.
>
> Once again, thank you for volunteering your time and effort to review our paper. We have incorporated your suggestion of not making claims sounding overly certain.
>
> We would greatly appreciate it if you could re-evaluate our paper considering the clarifications we have provided.
>
> Sincerely,
>
> Authors

---

> ### Author Response · Authors · 2024-11-23
> **Appreciating Your Feedback**
>
> We would appreciate your feedback on if any concerns remain as the discussion phase comes to a close. Thank you for your time and we are grateful for the discussion opportunity.
>
> Thanks!

---

### Official Review · Reviewer_ayAW · 2024-11-04

**Soundness:** 2
**Presentation:** 1
**Contribution:** 3
**Rating:** 3
**Confidence:** 4

**Summary:**

The paper works on the dual problem of the transport problem of multi-draft speculative decoding and show that the optimal acceptance rate is equivalent to a subset selection problem. Then, the paper provides several methods to compute such rates for commonly used multi-draft proposal methods (sampling with replacement, sampling without replacement). The paper proposes a greedy draft construction method and provides several empirical results that showcase the benefit of the proposed algorithm.

**Strengths:**

It is interesting to see that two existing verification approaches (K-Seq and the widely used RRS) can be unified as solving the same optimal transport problem coresponding to sampling without replacement $p_{draft}$. Therefore, they share the same upper bound. Table 1 shows that for a variety of models and settings, the two methods are close enough to the optimal acceptance rates.

The proposed "greedy" draft generation approach and verification method is an interesting combination of the greedy decoding method and ordinary sampling method.

The ablation studies are informative with a clear comparison with other baseline methods across different temperature and number of draft tokens.

**Weaknesses:**

### significance

Much portion of the paper is dedicated to theoretical derivations of the optimal acceptance rate. However, the description and the development of the proposed algorithm is underplayed.

The proposed methods deserve a proper name, clear demonstration of the verification algorithm (is the algorithm practical for $n>2 as compared to SpecHub?) and more thorough theoretical and experimental investigations to demonstrate the pros and cons compared with previous algorithms.


### clarity

The clarity of the paper can be significantly improved, including but not restricted to:

(1). In Section 2.2, the informal description of speculative decoding is very confusing. This seems to be a short summary of the formal description below, but it is hard to understand what $\max P(i=j)$ means before going into the details of the optimal transport problem.

(2). The same problem also applies to Section 2.3, where the informal description of multi-draft speculative decoding is confusing. I would suggest reframing the descriptions and move the examples of $p_{draft}$ after the definition.

(3). Section 3 is dedicated to provide the proof for the subset selection problem (Eq. 8). I would suggest move the proof details to the appendix and optionally write a short proof sketch section that only displays the important idea behind the proof and/or the part of the proof that is needed for the development of the later sections.

(4). The description of Theorem 3 and 4 can be improved. What is the definition of $Q$ and $q$ in these cases? What are the consequences of the special cases (with replacement and without replacement)?

(5). In Table 1, are "greedy" and "verify" the proposed methods in Section 5.1 and 5.2?


----

Given the above review, I would recommend the authors to further refine the writing and the presentation of the paper and put more efforts into both the theoretical part and the empirical results that supports the proposed algorithms.

**Questions:**

N/A.

---

> ### Author Response · Authors · 2024-11-15
> **Author's Response (1)**
>
> Thank you for volunteering your time and effort to review our paper.
>
> Regarding significance:
>
> We will comment on each of your points below to clarify the misunderstandings.
>
> Regarding clarity:
>
> We have revised the paper to refine the writing and presentation without changing any of the results, in order to help you understand the article better. Could you please check if it is now clear enough to not affect your understanding of our contributions? We believe the only reliable metric for measuring clarity is whether the reader can understand the contributions of the paper. If there are still parts you cannot understand, please let us know so we can improve how we express them. Otherwise, if you feel you can now understand all the contributions of this paper, would you please consider revisiting your assessment of the clarity?
>
> We would greatly appreciate it if you could kindly re-evaluate our paper, taking into account the clarifications we have provided.
>
> Sincerely,
>
> Authors
>
> ---
> ### Regarding significance:
>
> > However, the description and the development of the proposed algorithm is underplayed.
>
> There might be a misunderstanding. This paper provides two novel and useful algorithms:
> 1. Section 4.2.1 reduces the time complexity of the subset selection problem in the paper from $2^{|\Sigma|}$ to $O(|\Sigma|\log|\Sigma|)$, a significant improvement achieved by deeply analyzing the problem structure and proposing the highly original q-convexity to guide algorithm design.
> 2. Section 5 proposes the greedy draft construction method, which improves the optimal acceptance rate and comes with a verification algorithm that achieves the optimal acceptance rate.
>
> > The proposed methods deserve a proper name, clear demonstration of the verification algorithm
>
> We apologize that the name doesn’t fully convey the novelty of our approach. We are open to suggestions for a more suitable name if you have any recommendations.
>
> We note that the name does not change our four contributions shown in the introduction, which provide multiple new theoretical insights and new algorithms.
>
> We have also unfolded the math definition in the verification algorithm, hoping it will help you understand better.
>
> > is the algorithm practical for $n>2 as compared to SpecHub?
>
> By definition, SpecHub does not support the case of $n>2$, as we emphasized on line 335. Therefore there is no way to compare.
>
> > more thorough theoretical and experimental investigations to demonstrate the pros and cons compared with previous algorithms.
>
> Our contributions contain rigorous mathematical proofs, many novel theorems, and extensive experimental verification. We have achieved very small error bars, as can be seen in the nearly invisible error ranges in the tables and figures. Therefore the signal in our experiment is very strong and the noise very small.  We would be grateful if you could share what specific additional experiments or theoretical investigations would help strengthen the paper in your opinion, and what additional research questions you would like us to explore.
>
> > It is interesting to see that two existing verification approaches (K-Seq and the widely used RRS) can be unified as solving the same optimal transport problem coresponding to sampling without replacement $p_{draft}$. Therefore, they share the same upper bound.
>
> This is not our contribution and we did not claim it as such. This is a previously known result - in the SpecTr paper proposing K-Seq, the authors already pointed out this unifying view, so the credit for this contribution should go to them.
>
> Our contributions are:
> 1. Theoretical upper bound by transforming the problem of solving the optimal acceptance rate corresponding to the optimal transport into a subset selection problem
> 2. Solving the subset selection if the draft distribution satisfies certain properties
> 3. Measuring the theoretical upper bound of MDSD efficiency on real text, and the gap of existing verification algorithms
> 4. New greedy Multi-Draft Speculative Sampling algorithm that improves the theoretical upper bound in many situations.

---

> ### Author Response · Authors · 2024-11-15
> **Author's Response (2)**
>
> ### Regarding clarity:
>
> In our revision, we have refined the writing and presentation of the paper to address all the points below without changing any of the results, in order to help reviewers understand the article.
>
> Here are point-by-point responses:
>
> > (1). In Section 2.2, the informal description of speculative decoding is very confusing. This seems to be a short summary of the formal description below, but it is hard to understand what $\max P(i=j)$ means before going into the details of the optimal transport problem.
>
> We added a natural language explanation: "that is to maximize the probability of random variable $i$ to be the same as random variable $j$."
>
> > (2). The same problem also applies to Section 2.3, where the informal description of multi-draft speculative decoding is confusing. I would suggest reframing the descriptions and move the examples of $p_{draft}$ after the definition.
>
> We added a natural language explanation: "that is to maximize the probability of random variable $i$ to be the same as one of random variable in $(\bar{i}_1,\dots,\bar{i}_n)$."
> We did not move content up and down because the essential content is the same.
>
> > (3). Section 3 is dedicated to provide the proof for the subset selection problem (Eq. 8). I would suggest move the proof details to the appendix and optionally write a short proof sketch section that only displays the important idea behind the proof and/or the part of the proof that is needed for the development of the later sections.
>
> The current proof already only shows the important ideas, with details left to the appendix, such as Lemma 1 and Appendix C.1, just as you envisioned. We have highlighted the final conclusion at the beginning of Section 3. Readers uninterested in the proof are free to skip it, while interested readers can refer to it. This has been clarified in the original text.
>
> > (4). The description of Theorem 3 and 4 can be improved. What is the definition of $Q$ and $q$ in these cases? What are the consequences of the special cases (with replacement and without replacement)?
>
> $Q$ and $q$ are well-defined:
> - $Q$:
>   - line 250: $Q(H)=\sum_{\overline{i}\in H^n}p_{\mathrm{draft}}(i)$
>   - lines 156-161:
>     - Sampling with replacement: $p_{\mathrm{draft}}(\bar{i})=\prod_{j=1}^{n}q(\bar{i}_{j})$
>     - Sampling without replacement: $p\_{\mathrm{draft}}(\overline{i})=\prod\_{j=1}^nq^{\neg\overline{i}\_1,\ldots,\overline{i}\_{j-1}}(\overline{i}\_j)$, where $q^{\neg\overline{i}\_1,...,\overline{i}\_{j-1}}(x)=\begin{cases}\frac{q(x)}{1-\sum_{z\in\{\bar{i}\_1,\ldots,\bar{i}\_{j-1}\}}q(z)}&x\notin\{\bar{i}\_1,\ldots,\bar{i}\_{j-1}\},\\\\ 0&x\in\{\bar{i}\_1,\ldots,\bar{i}\_{j-1}\}&\end{cases}$
> - $q$: output distribution of the draft model (line 156)
>
> We have added an explanation to clarify this in case readers missed it in the paper.
>
> > (5). In Table 1, are "greedy" and "verify" the proposed methods in Section 5.1 and 5.2?
>
> Yes.
>
> > Given the above review, I would recommend the authors to further refine the writing and the presentation of the paper
>
> We have added additional explanations. Could you please check if it is now clear enough to not affect your understanding of our contributions? We believe the only reliable metric for measuring clarity is whether the reader can understand the contributions of the paper. If there are still parts you cannot understand, please let us know so we can improve how we express them. Otherwise, if you feel you can now understand all the contributions of this paper, would you please consider revisiting your assessment of the clarity?
>
> ### Regarding soundness:
>
> The other three reviewers all recognize the soundness of this paper:
> - Reviewer q6tv: Soundness: 4: excellent
> - Reviewer WnL3: Paper is mathematically rigorous
> - Reviewer DHWn: rigorously give some theoretical findings, including the upper bound of MDSD and a efficient method to compute the theoretical acceptance rate.
>
> You did not comment on soundness, only stating "Soundness: 2: fair".
>
> We would greatly appreciate it if you could provide more details on the specific aspects that led to your assessment of the soundness, so we can better address your concerns.
>
> You didn't mention any error of our paper in your comment, therefore we are not aware of the basis of your evaluation of soundness. Only by knowing the basis of your evaluation of soundness can we have a targeted discussion to clarify any misunderstandings.

---

> ### Author Response · Authors · 2024-11-15
> **Author's Response (3)**
>
> ### Regarding request for more results:
>
> > put more efforts into both the theoretical part and the empirical results that supports the proposed algorithms.
>
> We have presented 4 contributions, as summarized in the Introduction section, which provide multiple new theoretical insights and new algorithms, in the current paper. What additional research questions would you like us to explore? Please let us know.
>
> ### Questions:
>
> Thank you for volunteering your time and effort to review our paper.
> We would be grateful for any further response you can provide to these following questions, as only by understanding the reviewer's specific expectations and viewpoint can we make targeted efforts to facilitate discussion.
>
> Based on our explanations:
> - If there are still parts you feel you don't understand, could you please let us know?
> - If you now feel you can understand all the contributions, would you please consider revisiting your assessment of the clarity?
>
> Could you please share what is the basis of your evaluation of the soundness of this paper?
>
> Could you please share what additional research questions you would like us to explore?
>
> ### Summary:
>
> Once again, thank you for volunteering your time and effort to review our paper. We have incorporated your suggestions to improve clarity.
>
> We would greatly appreciate it if you could kindly re-evaluate our paper, taking into account the clarifications we have provided.
>
> Sincerely,
>
> Authors

---

> ### Author Response · Authors · 2024-11-23
> **Appreciating Your Feedback**
>
> We would appreciate your feedback on if any concerns remain as the discussion phase comes to a close. Thank you for your time and we are grateful for the discussion opportunity.
>
> Thanks!

---

> > ### Comment · Reviewer_ayAW · 2024-12-01
> > **Thank you for the rebuttal**
> >
> > Dear authors,
> >
> > I have gone through your rebuttal and all the other reviewers' opinions on the paper. Unfortunately, I cannot increase my score. Two specific reasons are:
> >
> > **Significance**:
> >
> > 3 out of 4 main contributions are about the theoretical upper bounds of the acceptance rate of multi-draft SD algorithms. The developed theory and the proposed efficient algorithm for solving the theoretical upper bounds seem to be interesting.
> >
> > However, this is a higher-order problem whose significance depends on the effectiveness of MDSD. As is also mentioned by Reviewer DHWn, the MDSD framework does not necessarily lead to a higher throughput. There are still gaps between the **acceptance rate** and the final **speedup**/**throughput**. Obtaining a higher acceptance rate does not always lead to a higher speedup/throughput, and other effects like longer draft time need to be considered.
> >
> > I would like to see if the developed theory can be of **independent interest**, especially outside the field of MDSD. If the developed duality, q-convexity theory, and the proposed efficient algorithm can be applied to another empirical setting (maybe discrete optimal transport?) then this will strengthen the significance of the theoretical results.
> >
> > **Clarity**:
> >
> > The clarity of the paper can still be significantly improved even for the revised draft. For example:
> >
> > - Section 2.1 introduces the background of speculative decoding. The points 3 & 4 are hard to read and understand.
> > - For section 2.2, I would suggest deleting Lines 121-123, which is extremely hard to understand without reading the further descriptions of the optimal transport formulation --- maybe bring up OT first?
> > - Section 2.3 (multi draft) is the extension of section 2.2 (single draft), I would suggest introducing MDSD formally first, and writing the single draft setting as a special case. Besides, the overview of RSS, K-Seq, or SpecTr should be introduced somewhere in the main text.
> > - TOTAL UNIMODULARITY should not be the subsection name of section 3.2. This should not even be an independent subsection. It serves as a comment in the proof sketch, which can be moved to the appendix (full proof) without hurting the readability of the main text.
> > - Section 4 introduces the concept of q-convex function. This section will benefit from having more illustrating examples of the q-convex functions. E.g., when does it reduce to the normal convex function that we are more familiar with? What are the definitions of "supermodular functions" for readers who are not familiar with the context?
> > - Can you give one counter-example that a q-convexity function is not necessarily supermodular?
> >
> > Based on these considerations, I will respectfully maintain my score.

---

> > > ### Author Response · Authors · 2024-12-03
> > > **Author's Response (1)**
> > >
> > > Thank you for your feedback. We appreciate you taking the time to review our paper again and provide constructive comments. We have carefully considered your points and made revisions to carry out your suggestions. Below, we address each of your points:
> > >
> > > > However, this is a higher-order problem whose significance depends on the effectiveness of MDSD. As is also mentioned by Reviewer DHWn, the MDSD framework does not necessarily lead to a higher throughput. There are still gaps between the acceptance rate and the final speedup/throughput. Obtaining a higher acceptance rate does not always lead to a higher speedup/throughput, and other effects like longer draft time need to be considered.
> > >
> > > Our work brings real speedup and creates practical value, as shown in Table 2. You mentioned that Reviewer DHWn raised the point about gaps between the acceptance rate and the final speedup/throughput. However, we have not observed this gap in our experiments, which demonstrate genuine acceleration.
> > >
> > > This is consistent with other recent works. For example, in Table 7 of the EAGLE paper, both speedup and throughput improved:
> > >
> > > | Model | Batch size 1 | Batch size 2 | Batch size 3 | Batch size 4 | Throughput |
> > > |---------|-------------|-------------|-------------|-------------|------------|
> > > | Vicuna 7B | 2.90x | 2.87x | 2.65x | 2.76x | 1.97x |
> > > | LLaMA2-Chat 70B | 3.01x | 2.81x | 2.50x | 2.40x | 1.99x |
> > >
> > > The recent work "MagicDec: Breaking the Latency-Throughput Tradeoff for Long Context Generation with Speculative Decoding" also reported similar results, as we discussed in our response to Reviewer DHWn.
> > >
> > > In summary, our work is significant because it not only provides theoretical insights but also delivers practical performance improvements.
> > >
> > > > I would like to see if the developed theory can be of independent interest, especially outside the field of MDSD.
> > >
> > > While our theory may hold independent research significance in other related fields—a topic we are open to exploring further in the future—such considerations fall beyond the scope of our current study.
> > >
> > > Within MDSD, our contributions are already sufficiently significant, providing real acceleration and theoretical insights.
> > >
> > > > If the developed duality, q-convexity theory, and the proposed efficient algorithm can be applied to another empirical setting (maybe discrete optimal transport?) then this will strengthen the significance of the theoretical results.
> > >
> > > Our novel theory was not proposed for general discrete optimal transport problems. Instead, we analyzed the special class of discrete optimal transport problems in MDSD and proposed a theory and efficient algorithm tailored to the MDSD scenario. This is non-trivial and conveys a deep understanding of this class of problems.
> > >
> > > We agree that if future research finds broader applications of our novel concepts, it will further enhance their significance.
> > >
> > > However, we emphasize that the existing contributions in MDSD, improving real generation speed and providing theoretical insights, are already sufficiently significant.

---

> ### Author Response · Authors · 2024-12-03
> **Author's Response (2)**
>
> Regarding improving clarity, we have made revisions based on your suggestions:
>
> > Section 2.1 introduces the background of speculative decoding. The points 3 & 4 are hard to read and understand.
>
> We provided additional explanations. The draft tokens generated in speculative decoding need to be verified. Such a process is reflected in point 3. The verification result is $x_{m+1}$, and the verification process has randomness: $x_{m+1}\sim P_{\operatorname{verify}}(\cdot|\widehat{x}^{(1)},\dots\widehat{x}^{(n)})$. The specific verification result probability distribution depends on different algorithm implementations, such as eq (5) for single draft and eq (19) for our greedy method.
>
> If a draft token is accepted, it brings acceleration, as reflected in point 4. Multiple tokens can be generated in one forward pass, accelerating compared to basic autoregressive sampling, which generates a single token for each forward pass. Minimizing the forward process of large language models leads to acceleration.
>
> > For section 2.2, I would suggest deleting Lines 121-123, which is extremely hard to understand without reading the further descriptions of the optimal transport formulation --- maybe bring up OT first?
>
> Thank you for this valuable suggestion. We have deleted Lines 121-123 and directly provided the rigorous optimal transport formulation.
>
> > Section 2.3 (multi draft) is the extension of section 2.2 (single draft), I would suggest introducing MDSD formally first, and writing the single draft setting as a special case.
>
> Following your suggestion, we adjusted the order of Sections 2.2 and 2.3 and introduced single draft setting as a special case of general MDSD.
>
> > Besides, the overview of RSS, K-Seq, or SpecTr should be introduced somewhere in the main text.
>
> We have added brief overviews of these methods in at the end of Section 2.PRELIMINARIES to provide context:
>
> - Recursive Rejection Sampling (RRS): RSS (Yang et al., 2024b; Jeon et al., 2024) is one of the foundational multi-draft speculative decoding methods. It generates multiple draft tokens simultaneously and verifies them in a sequential manner, serving as a baseline for MDSD.
> - SpecTr and K-Seq Methods: The SpecTr paper (Sun et al., 2024d) pointed out that optimal acceptance rate is characterized by an optimal transport problem, and proposed K-Seq method to improve the acceptance rates both theoretically and practically.
>
> > TOTAL UNIMODULARITY should not be the subsection name of section 3.2. This should not even be an independent subsection. It serves as a comment in the proof sketch, which can be moved to the appendix (full proof) without hurting the readability of the main text.
>
> Following your suggestion, we removed the subsection name and only used a comment to explain that this is a step in the proof. The main content was moved to the appendix, keeping only a brief proof sketch in the main text to make room for your other suggested modifications.
>
> > Section 4 introduces the concept of q-convex function. This section will benefit from having more illustrating examples of the q-convex functions. E.g., when does it reduce to the normal convex function that we are more familiar with?
>
> The two do not overlap. A q-convex function is defined on subsets of a set $\Sigma$, where the input variable is a set. In contrast, the input variable of a convex function is a real number or vector. Therefore, a q-convex function does not reduce to a convex function.
>
> However, q-convex functions are inspired by convex functions, hence the name. We introduced the connection after Definition 2. For any ordering of the elements in $\Sigma$, we denote it as $\Sigma=\{\sigma_1,\dots,\sigma_{|\Sigma|}\}$, we can construct a sequence of sets $H_i$ by adding elements one by one: $H_i=\{\sigma_1,\dots,\sigma_{i}\}$. We can then construct a set of points $\{(x_i,y_i)\}_{1\leq i\leq|\Sigma|}$ in a two-dimensional plane, where $x_i=\sum_{\sigma\in H_i}q(\sigma)$ and $y_i=Q(H_i)$. Connecting these points with straight lines always forms a convex function.
>
> > What are the definitions of "supermodular functions" for readers who are not familiar with the context?
>
> Thank you for the suggestion. We have added the definition of supermodular functions: A function $f$ defined on subsets of a set $\Sigma$ is supermodular if for any two subsets $A\subseteq \Sigma$ and $B\subseteq \Sigma$, $f(A) + f(B) \leq f(A \cup B) + f(A \cap B)$.
>
> > Can you give one counter-example that a q-convexity function is not necessarily supermodular?
>
> As proved in Theorem 5, all q-convex functions are supermodular functions. Therefore, there is no counter-example.

---

> ### Author Response · Authors · 2024-12-03
> **Author's Response (3)**
>
> We have made modifications point by point in response to your comments above. Unfortunately, we have been unable to upload a new version of the PDF since Nov 27, but the changes will be reflected in the final version. We also emphasize that our work has significant practical implications, as it not only advances theoretical understanding but also delivers real acceleration in MDSD. Thank you again for your constructive feedback and the opportunity to improve the clarity of our paper.

---

### Official Review · Reviewer_q6tv · 2024-11-05

**Soundness:** 4
**Presentation:** 4
**Contribution:** 4
**Rating:** 8
**Confidence:** 3

**Summary:**

The paper presents several new results about Multi-Draft Speculative Sampling.
1. It transforms the problem of computing the optimal acceptance rate into a subset selection problem.
2. For some cases it provides a practical solution of the problem. This provides a theoretical upper bound on the acceptance rate.

The authors then measure the theoretical upper bound on some datasets, and measure the gap between the upper bound and previous algorithms on these datasets.
They present a greedy algorithm which is able to match the theoretical upper bound in many cases.

**Strengths:**

The paper makes progress on understanding the acceptance rate of Multi-Draft Speculative Sampling. The authors show a clever transformation of the transportation problem formulation of optimal acceptance rates to a subset selection problem, and then show an algorithm to solve the subset selection if the draft distribution satisfies certain properties.
They then propose a new greedy Multi-Draft Speculative Sampling algorithm, which is closer to the optimal acceptance rate on some datasets.

The results in the paper seem to me to be quite novel and significant.

**Weaknesses:**

The paper takes a bit of effort to read, partly because of a lot of notation, and partly because of results that may not be familiar to a lot f readers. I am not sure if the authors can do much about this.

**Questions:**

Do your results extend to trees of drafts in a straightforward way?

I would like to see a comparison to https://openreview.net/forum?id=N1L5TgtkAw

---

> ### Author Response · Authors · 2024-11-15
> **Author's Response**
>
> We sincerely appreciate the time and effort you have invested in reviewing our paper and providing valuable feedback. We are delighted to know that you find our contributions to be "quite novel and significant."
>
> To address your questions:
>
> > Do your results extend to trees of drafts in a straightforward way?
>
> Yes, our method has been extensively validated on trees of drafts with various structures, including k-branch trees and sparse tree structures, as shown in Table 2. Specifically, we implemented our algorithm based on the Eagle framework and tested its effectiveness on several configurations:
> - #Drafts = 2, #Steps = 4
> - #Drafts = 4, #Steps = 3
> - EAGLE default sparse tree
>
> The results in Table 2 demonstrate that our method performs consistently well across these scenarios, highlighting its robustness and adaptability to diverse tree structures.
>
> > I would like to see a comparison to https://openreview.net/forum?id=N1L5TgtkAw
>
> There is an interesting connection between our work and the mentioned paper "Multi-Draft Speculative Sampling: Canonical Architectures and Theoretical Limits." In Remark 2 of their paper, they conjecture that "the optimal acceptance probability in the general case of $K > 2$ drafts is attained by replacing the exponent of 2 in the second term in (9) to K." Our results not only prove this conjecture but are also more general, as their work only considers sampling with replacement, while our results apply to any draft distribution and any number $K$ of drafts.
>
> In other words, our first contribution is more general than Theorem 3 and Remark 2 in that mentioned paper.
>
> Our second and third contributions go further beyond their work:
>
> 2. Solving the subset selection problem if the draft distribution satisfies certain properties
> 3. Measuring the theoretical upper bound of MDSD efficiency on real text and the gap of existing verification algorithms
>
> We introduce the concept of q-convexity, which enables efficient and exact solutions to the subset selection problem. To our understanding, their work does not propose a structure similar to q-convexity, and thus cannot achieve efficient solutions. Even if they assume their Remark 2 holds, the subset selection problem has $2^{50272}$ possible variable assignments for the OPT model with $|\Sigma|=50,272$. Brute-force search would be even more costly than linear programming, which already requires petabyte-level storage for either our formulation ($C{\in}\mathbb{R}^{\Sigma\times\Sigma^n}$) and theirs ($\beta_y(x_{1:K}), \forall x_{1:K}\in\Omega^K, \forall y{\in}\Omega$). However, they still report the optimal acceptance probability in Table 2 for $K=2,4,8$. We are not sure how they computed these results as we could not find the exact information in their paper, and they did not provide code.
>
> Our unique methods, unlocked by a deeper understanding of the problem's structure, are far more efficient and can handle cases with tens of thousands of tokens without any approximation.
>
> Regarding practical methods, the unique contribution of that mentioned paper is a new verification method that more closely approximates the theoretical upper bound of sampling with replacement. Our fourth contribution takes a different path:
>
> 4. We propose a new greedy Multi-Draft Speculative Sampling algorithm that goes beyond sampling with replacement by considering new draft distributions and directly improving the theoretical upper bound.
>
> Our greedy draft construction produces a higher acceptance rate than the theoretical upper bound of sampling with replacement.
>
> We understand that the paper may require some effort to read due to the notation. We have added an extra section in Appendix E, summarizing all the notations, acting as a handy reference for readers.
>
> Once again, we express our gratitude for your time and your recognition of our work.
>
> Sincerely,
>
> Authors

---

> > ### Author Response · Authors · 2024-11-23
> > **Appreciating Your Feedback**
> >
> > Thank you for your time and recognition of our novel contribution. We would appreciate your feedback on if any concerns remain as the discussion phase comes to a close. We noticed that the other paper you invited for comparison also provided one, and we wanted to clarify a minor inaccuracy on "The experiments reported in the other paper appear to be limited to a specific draft sampling scheme": our experiments considered multiple sampling schemes, including with/without replacement, SpecHub and a new greedy draft distribution. Nevertheless, we believe both papers make unique contributions. We are grateful for this discussion opportunity.

---

> > > ### Comment · Reviewer_q6tv · 2024-11-24
> > > **Thank you for the comparison.**
> > >
> > > Please include it in the final version of your paper.

---

### Meta-Review · Area_Chair_2gDE · 2024-12-23

**Metareview:**

This paper studies multi-draft speculative decoding and provides multiple theoretical contributions. The paper starts with the optimal transport formulation of multi-draft speculative decoding acceptance rate (Sun et al., 2023) and derives its dual. Then it is shown that the optimal solution to the dual problem takes a subset selection form. Then, solutions are obtained under mild assumptions for three key drafting strategies: (1) sampling with replacement; (2) sampling without replacement; and (3) so-called greedy decoding (which I refer to as almost-top-k sampling as the name greedy already has a meaning and should not be overloaded) where top-(k-1) tokens are selected and then the last token is selected by sampling without replacement. Several key insights are derived in the sequel. The case of k=2 degenerates to concurrent work of (Khisti et al., 2024) that fully solves the optimal transport problem for k=2 and the same acceptance rate is derived. There is a large gap between sampling with replacement and sampling without replacement. The proposed almost-top-k sampling (called greedy in the paper) offers further improvements over sampling without replacement, which can be substantial in some regimes. The findings are corroborated through experiments with several draft and verification models. The reviewers' main concern is that the impact of multi-draft speculative decoding is limited in cases where large batch sizes are used due to throughput limitations. Also, the reviewers note that the applicability of the theory in this paper beyond speculative decoding is unclear. While the AC agrees with these weaknesses, I think the theoretical contributions of this paper are non-trivial on their own right and help further understand the theoretical limits of multi-draft speculative decoding. As such, the paper is recommended to be accepted. Congratulations to the authors!

Sun, Ziteng, et al. "Spectr: Fast speculative decoding via optimal transport." Advances in Neural Information Processing Systems 36 (2023).

Khisti, Ashish, et al. "Multi-Draft Speculative Sampling: Canonical Architectures and Theoretical Limits." arXiv preprint arXiv:2410.18234 (2024).

**Additional Comments On Reviewer Discussion:**

While the reviewers have recommended the paper to be rejected, after a careful reading of the paper and further consultation with an additional expert reviewer and the SAC, the AC believes that the paper makes a significant contribution that deserves to be accepted in the conference. In addition, the concerns of the reviewers (while correct) appear not to be a blocker for publication.

---

### Decision · Program_Chairs · 2025-01-22

Accept (Poster)